# CD206+ Trem2+ macrophage accumulation in the murine knee joint after injury is associated with protection against post-traumatic osteoarthritis in MRL/MpJ mice

Jillian L. McCool [1,2], Aimy Sebastian[1], Nicholas R. Hum[1], Stephen P. Wilson[1], Oscar A. Davalos[1], Deepa K. Murugesh[1], Beheshta Amiri[1], Cesar Morfin[1,3], Blaine A. Christiansen [3], Gabriela G. Loots [1,2,3]*

1 Lawrence Livermore National Laboratory, Physical and Life Science Directorate, Livermore, CA, United States of America, 2 School of Natural Sciences, University of California Merced, Merced, CA, United States of America, 3 Department of Orthopaedic Surgery, University of California Davis Health, Sacramento, CA, United States of America

* gloots@ucdavis.edu

**Data Availability Statement:** The datasets generated for this study can be found in the Gene

## Abstract

Post-traumatic osteoarthritis (PTOA) is a painful joint disease characterized by the degradation of bone, cartilage, and other connective tissues in the joint. PTOA is initiated by trauma to joint-stabilizing tissues, such as the anterior cruciate ligament, medial meniscus, or by intra-articular fractures. In humans, ~50% of joint injuries progress to PTOA, while the rest spontaneously resolve. To better understand molecular programs contributing to PTOA development or resolution, we examined injury-induced fluctuations in immune cell populations and transcriptional shifts by single-cell RNA sequencing of synovial joints in PTOA-susceptible C57BL/6J (B6) and PTOA-resistant MRL/MpJ (MRL) mice. We identified significant differences in monocyte and macrophage subpopulations between MRL and B6 joints. A potent myeloid-driven anti-inflammatory response was observed in MRL injured joints that significantly contrasted the pro-inflammatory signaling seen in B6 joints. Multiple CD206+ macrophage populations classically described as M2 were found enriched in MRL injured joints. These CD206+ macrophages also robustly expressed *Trem2*, a receptor involved in inflammation and myeloid cell activation. These data suggest that the PTOA resistant MRL mouse strain displays an enhanced capacity of clearing debris and apoptotic cells induced by inflammation after injury due to an increase in activated M2 macrophages within the synovial tissue and joint space.

## Introduction

Osteoarthritis (OA) is a painful joint disease that affects over 300 million people world-wide and is a leading cause of disability. Post-traumatic osteoarthritis (PTOA) is a form of OA that develops as a result of joint injuries such as anterior cruciate ligament (ACL) rupture or

Expression Omnibus (GEO) under accession numbers GSE200843 and GSE220167.

**Funding:** Department of Defense Awards PR180268 (GGL), PR180268P1 (BAC) and PR192271 (AS). Lawrence Livermore National Laboratory grant LDRD 20-LW-002 (AS) National Institute of Health grant R01 AR075013 (BAC). The funders had no role in study design, data collection and analysis, decision to publish, or preparation of the manuscript.

**Competing interests:** The authors have declared that no competing interests exist.

meniscal tear and accounts for ~12% of all OA cases [1]. This disease has a high cost burden to healthcare systems worldwide and in the United States alone exceeds $10 billion annually [2]. Currently, there are no approved therapies to prevent the chronic pain and joint dysfunction associated with PTOA. Therefore, more research is needed to understand the molecular mechanisms of PTOA pathogenesis and to identify new targets for therapeutic development.

In humans, nearly half of traumatic knee joint injuries progress to PTOA, while the remainder spontaneously resolve without progressive cartilage degeneration, independent of whether corrective surgery occurs. The pathogenesis and onset of PTOA are still not fully understood, but multiple factors such as genetics, epigenetics, and immune responses have been implicated in disease progression [3]. This study examined the immune system's responses to ACL injury in C57BL/6J (B6), a PTOA vulnerable mouse strain and 'super-healer' MRL/MpJ (MRL) mice, a strain resistant to PTOA [4] to better understand the immune-driven mechanisms of resistance to joint degeneration. The MRL strain has been described to have an extraordinary capacity for regenerating soft tissues after damage, as well as repairing injured cartilage [4–12]. In a noninvasive tibial compression injury model, our group previously showed that MRL knee joints remain resistant to cartilage degradation for at least 12 weeks post injury, the latest time point examined [11]. These injured MRL joints also developed significantly less osteophytes and displayed comparable OARSI scores to uninjured controls, indicating a non-arthritic joint resolution post injury [11]. While the ability of MRL to repair their knee joint has been histologically evaluated in several studies [5, 12–14] and decreased inflammation and increased M2 macrophage polarization have been suggested to contribute to the PTOA resistance [14, 15], the precise cellular and molecular interactions leading to this resistant phenotype have not yet been fully elucidated. Identifying the cellular and molecular mediators of PTOA resistance in MRL could open up new avenues for therapeutic development for PTOA.

The timeline of PTOA progression following joint injury can be classified into several phases starting at trauma. After the immediate sequelae of injury, an acute/subacute phase, dominated by inflammation, leukocyte infiltration, and tissue remodeling occurs. This inflammatory phase can spontaneously resolve after a few weeks or months, or progress to a chronic phase that can last for years, during which metabolic changes in the tissue progress through a clinically asymptomatic period that eventually leads to PTOA with severe joint pain and restricted mobility [16] that may become debilitating without surgical intervention [17]. In this study, we focused on the acute/subacute phase to determine the immune changes that could contribute to PTOA susceptibility or resistance.

An increasing number of studies ranging from horses to mice have shown that arthritis progression is dependent on the immune system's response to injury [18–22]. These studies have implicated resident and infiltrating immune cell types including macrophages, monocytes, neutrophils, dendritic cells, B and T cells in the pathogenesis of osteoarthritis of synovial joint tissue [1, 23, 24]. Macrophages are the major immune cell type present in healthy synovial tissues of the joint; they are essential in maintaining the integrity of the synovial cavity to keep articular cartilage unperturbed by endogenous damage-associated molecular patterns (DAMPs) that form from wear and tear of the joint [25]. During acute inflammation, as in the case of injury, there is an increase in monocytes, activated macrophages, and synovial fibroblasts that enter the joint space due to a disruption of the synovial lining [26, 27]. This influx leads to an expansion of the synovial pannus and degradation of the articular cartilage due to a spike in metalloproteinases secreted by infiltrating monocyte-derived macrophages [28]. Some subpopulations such as the $Trem2^+$ (Triggering Receptor Expressed On Myeloid Cells 2) alternatively activated macrophages have been described as anti-inflammatory and are likely to promote healing and repair of damaged tissues [29–31]. Previously, we identified $Trem2^+$ macrophages as a major subpopulation in B6 mice that expands in response to knee injury [32].

Characterizing key cell types like Trem2[+] macrophages that prompt an anti-inflammatory phenotype is crucial in understanding immune cell function that aids healing and prevents PTOA development, post injury. Additionally, these subpopulations have clinical relevance as potential cell-based therapies where macrophages of appropriate phenotypes can directly improve healing or enable the production of macrophage-derived therapeutic proteins for long term damage control [33, 34].

To enhance our understanding of the role the immune system plays in joint repair or joint degeneration after injury, we employed single-cell RNA sequencing (scRNA-seq) analysis of CD45[+] cells from injured and uninjured knee joints of ~~from~~ PTOA-resistant MRL and vulnerable B6 mice by using a noninvasive tibial compression injury model that mimics ACL rupture and subsequent PTOA development in humans [11, 35–38]. This study highlights significant differences in myeloid subpopulations in the synovial capsule and infrapatellar fat pad of MRLs after injury including increased presence of anti-inflammatory Trem2[+] macrophages and reduced expression of pro-inflammatory genes. In addition, we identified a number of genes highly expressed in immune cells from MRL compared to B6 including Glyoxalase 1 (*Glo1*) and Von Willebrand Factor A Domain Containing 5A (*Vwa5a*), which may play a role in PTOA resistance observed in MRL. The immune characterization of 'PTOA-resistant' MRL and 'PTOA-vulnerable' B6 joints presented here identified several strain-specific differences that correlate with a disease protection phenotype and should be further explored mechanistically and therapeutically.

## Materials and methods

### Experimental animals and ACL injury model

MRL (MRL/MpJ, Stock # 000486) and B6 (C57BL/6J, Stock # 000664) animals were purchased from Jackson Laboratory and bred in house using standard procedures. Ten-week-old male MRL and B6 were anesthetized using isoflurane inhalation and subjected to non-invasive, non-surgical, knee joint injury as previously described [39]. Briefly, the right lower leg was placed between two platens and was subjected to single tibial compression overload (~10-16N) at 1 mm/s displacement rate to induce an ACL rupture using an electromagnetic material testing system (ElectroForce 3200, TA Instruments, New Castle, DE, USA). Mice were administered a 50 μL dose of 0.9% sterile saline (Becton, Dickinson and Company, Franklin Lakes, NJ, USA), and a body mass dependent dose of buprenorphine (0.01 mg/kg) immediately post-injury for pain relief. Mice were then allowed normal cage activity while on 12h light/dark cycles prior to euthanasia at terminal time points. All animal experimental procedures were completed in accordance with the Institutional Animal Care and Use Committee (IACUC) guidance at Lawrence Livermore National Laboratory and the University of California, Davis in AAALAC-accredited facilities.

### Histological assessment of the articular joint

After ACL injury, right hindlimbs (n = 5/group) were collected from uninjured day 0 (D0) and injured mice at day 7 (D7) and 4 weeks (4W) post-injury and processed for histological evaluation as previously described [32]. Briefly, whole hindlimbs were fixed in 10% Neutral Buffered Formalin (NBF), decalcified using 0.5 M EDTA using the weight loss-weight gain method for measuring decalcification status [40] and processed for paraffin embedding. Joints were sectioned in the sagittal plane at 6 μm and serial medial sections were prepared for histological assessment of joint tissue integrity at all timepoints. Sections were stained on charged glass slides using 0.1% Safranin-O (0.1%, Sigma, St. Louis, MO, USA; S8884) and 0.05% Fast Green (0.05%, Sigma, St. Louis, MO, USA; F7252) using standard procedures (IHC World,

Woodstock, MD, USA). Slides were imaged using a Leica DM5000 microscope (Leica Microsystems, Wetzlar, Germany). ImagePro Plus V7.0 Software, a QIClick CCD camera (QImaging, Surrey, BC, Canada), and ImageJ V1.53 Software were used for imaging and photo editing [41].

## OARSI histological scoring of joint degradation

Serial medial sections from B6 and MRL (n = 5/strain) were stained using Safranin-O and Fast Green as described above and subjected to a blinded semi-quantitative scoring by five individual scientists using the OARSI Histopathology Scoring System [42]. All scores were averaged, and mean score was plotted to determine the grade of joint damage that had occurred at 4W post injury.

## Immunohistochemistry

Serial medial sections from B6 and MRL (n = 5/strain) were subjected to antigen retrieval with Unitrieve (NB325 Innovex Biosciences, Richmond, CA. USA) and blocking using Background Buster (NB306 Innovex Biosciences, Richmond, CA. USA) per manufacturer's instructions. Samples were stained with primary antibodies and incubated overnight at 4°C in a dark, humid chamber. Samples were washed and incubated for 2 hours at room temperature in a dark, humid chamber with secondary antibodies at 1:500. Negative control slides were incubated with secondary antibody only. Stained slides were mounted with Prolong Gold with DAPI for nuclei staining (Molecular Probes, Eugene, OR. USA). Slides were imaged using a Leica DM5000 microscope. ImagePro Plus V7.0 Software, QIClick CCD camera (QImaging, Surrey, BC, Canada) and ImageJ V1.53 Software were used for imaging and photo editing. Primary antibodies included: Trem2 [1:100; ab95470 Abcam, Cambridge, UK], CD206 [1:100; ab64693, Abcam, Cambridge, UK], S100a8 [1:100; ab92331 Abcam, Cambridge, UK], Lyve1 [1:100; ab218535 Abcam, Cambridge, UK], Ly6G [1:100; ab238132 Abcam, Cambridge, UK]. Secondary antibodies included goat anti-rabbit 594 (1:1000; A11037, Thermofisher, Waltham, MA. USA), donkey anti-goat 488 (1:1000; A11055, ThermoFisher, Waltham, MA. USA).

## Single-cell RNA sequencing and data analysis

D0 (uninjured) and joints (n≥4/time point/strain) from day 1 (D1), D3, D7, 2 weeks (2W), and 4W post-injury were collected from MRL and B6 mice for scRNA-seq analysis. Mice were euthanized humanely under $CO_2$ and entire hindlimbs were dissected free of any superficial tissues such as the muscle, retaining the synovial fluid between the tibia and femur. To obtain immune cells from the joint without any bone marrow contamination, joint-residing cells from intact joints were released by digesting the soft tissues around the joint. Cells residing in the synovial capsule were collected by separating the joint between the femur and tibia into 7.5 mL of DMEM/F12 containing 3% Collagenase 1 solution (Worthington Biochemical, Lakewood, NJ; CLS-1) and 100 μg/mL DNase I (Roche, Basel, Switzerland; 11284932001). Hindlimbs with joint tissues were then digested while shaking at 37°C for two 1-hour digests and then filtered through a 100μm nylon cell strainer to remove large tissue fragments. After digestion, red blood cell lysis was performed with ammonium-chloride-potassium (ACK) lysis buffer (ThermoFisher Scientific, Waltham, MA, USA; A1049201) then CD45$^+$ immune cells were enriched using CD45-conjugated magnetic microbeads (Miltenyi Biotech, Bergisch Gladbach, Germany; 130-052-301) followed by Miltenyi Biotech MACS separation with LC columns. For bone marrow cell isolation, femur and tibia were first isolated from uninjured, 10-week-old male BL6 mice. The bones were then gently crushed to expose the marrow cavity and thoroughly rinsed with PBS until all the marrow was flushed out of the bone. Bone marrow cells were then pelleted, and ACK red blood cell lysis was performed. All final cell

preparations were resuspended in PBS with 1% FBS for scRNA-seq preparation. Each scRNA-seq sample was comprised of pooled 3–5 mouse replicates to mitigate biological variability. Immune (CD45$^+$) joint populations were sequenced using a Chromium Single Cell 3' V3 Reagent Kit and Chromium instrument (10x Genomics, Pleasanton, CA). Library preparation was performed according to the manufacturer's protocol and sequenced on an Illumina Next-Seq 500 (Illumina, San Diego, CA, USA).

Raw scRNA-seq data were processed using the Cell Ranger software (10x Genomics, Pleasanton, CA, USA) as described before [32]. Briefly, raw base call (BCL) files generated by Illumina NextSeq 500 sequencer were demultiplexed into FASTQ files using Cell Ranger 'mkfastq'. Aligning sequencing data to the mouse genome (mm10), barcode counting, and unique molecular identifier (UMI) counting were performed using Cell Ranger 'count'. Subsequently, rRaw count matrices generated with Cell Ranger were loaded into R (v4.3.2) and merged into a single object for downstream analysis using Seurat (v4.3.0) [43]. Low quality reads were filtered out and cells satisfying the following criteria were retained for further analysis: number of read counts $\geq$ 500; number of genes $\geq$ 200; mitochondrial gene percentage $<$ 10. Genes expressed in less than 10 cells were also removed. Data was then normalized using the 'NormalizeData'function with default parameters. Subsequently, tThree thousand highly variable genes (HVGs) were identified using the "vst" method. Before dimension reduction the data was scaled using only HVGs with following variables regressed out: number of read counts and mitochondrial percentage. After scaling, principal component analysis (PCA) was performed and principal components (PCs) 1–50 were used for subsequent analysis. Data integration i.e. batch correction was performed using Harmony with the grouping variable being "orig.ident" which contained all individual samples [44]. Clusters were identified using 'FindNeighbors'and 'FindClusters'with the reduction parameter set to "harmony" and resolution 0.2. A non-linear dimensional reduction was then performed *via* uniform manifold approximation and projection (UMAP) with the following parameter modifications: reduction = "harmony"; umap.method = "uwot"; spread = 4. Cluster marker genes were identified using 'FindAllMarkers'with the parameter only.pos set to true. Monocytes and macrophages (Mono/Mac; Csf1r$^+$ Itgam$^+$ cluster) and neutrophils (S100a8$^+$ S100a9$^+$ cluster) were all extracted and analyzed further following the same methods as above with the following differences: Mono/Mac (2000 HVGs, 1–40 PCs, resolution 0.5); Neutrophils (2000 HVGs, 1:40 PCs, resolution 0.5). Differential gene expression analysis between mouse strains was conducted by isolating the relevant cell type and applying 'FindAllMarkers'with only.pos set to true. For neutrophil subpopulations, gene ontology (GO) enrichment analysis was performed on up to 100 differentially expressed genes per cluster using clusterProfiler(v4.10.0) [45, 46]. For genes differentially expressed between MRL and B6 in specific macrophage subpopulations (log2FC $>$ 0.25; FDR $<$0.05), GO analysis was performed using ToppGene Suite [47] and enrichment dot plots were generated using custom R scripts. Pathway and transcription factor activity inferences were performed and visualized with decoupleR (v2.8.0) and SCpubr (v2.0.2) [48, 49]. All data wrangling and analysis was performed in R (v4.3.2) using tidyverse (v2.0.0) functions. Data visualization leveraged tools already mentioned above and a mixture of khroma (v1.11.0), ggthemes (5.0.0), and Rcolorbrewer (v1.1.3) for color palettes [50, 51].

Neutrophils from D0 B6 joints were also compared to those from D0 bone marrow (BM) isolates. BM immune scRNA-seq data was integrated with immune scRNA-seq data from D0 B6 joints using Seurat's anchor-based canonical correlation analysis (CCA) integration method. CCA integration was performed by identifying 2000 HVGs per dataset, followed by applying 'SelectIntegrationFeatures', 'FindIntegrationAnchors', and 'IntegrateData'functions using default parameters. After CCA based integration, data processing followed the steps previously described above.

## Single cell trajectory analysis

Single cell pseudo-time trajectories of immune cell subpopulations were constructed with Monocle [52]. Following analysis of scRNA-seq data in the Seurat object format; the expression data, phenotype data, and feature data were extracted for constructing Monocle's "Cell-DataSet" object utilizing the "newCellDataSet" function. Highly variable genes from within the Seurat object were selected as ordering genes. The Monocle "reduceDimension" function was used to reduce the dataset's dimensionality using the DDR algorithm. Ordering of cells along the computed trajectory was carried out using the "orderCells" function with default parameters.

## Perfusion of mice

For the perfusion of blood, mice were anesthetized by administering isoflurane (4–5% in 100% oxygen *via* a nose cone. Once mice were no longer responsive to tail pinch reflex, the thoracic cavity was opened through the diaphragm, and ribs were cut bilaterally to expose the heart. A butterfly needle was then inserted into the left ventricle and secured. Next, a small incision was made in the right atrium to create an outlet for effluent. With the aid of a perfusion pump (flow set at 10ml/min), mice were perfused with 20ml of sterile PBS + 0.1% heparin.

## Flow cytometry analysis

Single cell suspensions from injured and uninjured knee joints were generated as described above in the scRNA-seq section (n = 3-5/group). Cells were blocked using rat anti-mouse CD16/CD32 (Stock # 14-0161-82, Mouse Fc Block; Thermo Fisher, Waltham, MA. USA) at 4°C for 10 minutes then incubated with an antibody cocktail (Thermo Fisher) specific for macrophage characterization containing the following antibodies at a 1:100 dilution: PerCP CD45 monoclonal antibody (Clone: EM-05, Stock# MA110234), eFluor 506 CD11b monoclonal antibody (Clone: M1/70, Stock# 69-0112-82), PE F4/80 monoclonal antibody (Clone: QA17A29, Stock 157304), APC CD206/MMR monoclonal antibody (Clone: MR6F3, Stock# 17-2061-82), FITC TREM2 monoclonal antibody (Clone: 78.18, Stock# MA528223) and DAPI for viability staining. To identify proportions of myeloid cells, isolated cells from the knee joints at all timepoints were stained using Biolegend antibodies at 1:100 dilution: APC/Cy7 anti-mouse CD45 antibody (Clone: 30-F11), FitC anti-mouse/human CD11b antibody (Clone: M1/70), Brilliant Violet 510 anti-mouse Ly-6C (Clone: HK1.4), APC anti-mouse Ly-6G (Clone: 1A8) and DAPI for viability staining. Flow cytometry was also performed on perfused mice and neutrophil populations were identified within the joint cell suspensions using the following antibodies (BioLegend, San Diego, CA USA): anti-mouse APC CD45 antibody (Clone: 30-F11), anti-mouse FITC CD11b (Clone: M1/70), and anti-mouse APC/Cyanine7 Ly6g (Clone: 1A8) at a 1:100 dilution in PBS +1% FBS and DAPI was used as a viability stain. All flow cytometric analyses were performed on a BD FACSMelody system.

## Analysis software and statistical analysis

Statistical analyses were performed using GraphPad Prism (n = 3–5 biological replicates per strain). A one-way ANOVA and post-hoc Bonferroni's Test were used to assess statistically significant differences of mean expression values. OARSI scoring is presented from 4 biological replicates per strain and scored by 5 individual scientists. A one-way ANOVA and post-hoc Bonferroni's Test were used to assess statistically significant differences of mean expression values. All results were considered statistically significant for $p$-values <0.05.

## Results

### ScRNA-seq reveals differences in knee joint immune landscape after injury in MRL and B6 mice

Consistent with prior reports [8, 35, 37, 53], B6 mice showed visible proteoglycan loss, fibrillation, and significant erosion to the calcified cartilage layer by 4W post injury, while injured MRLs retained their pre-injury cartilage thickness with non-significant decreases in proteoglycan staining (**Fig 1A and 1B**). Single cell analysis of immune (CD45$^+$) cells from uninjured (D0), and D1, D3, D7, 2W, and 4W post injury joints identified changes in the immune profile of B6 and MRL mice before and after knee injury (**Fig 1C and 1D**). Seven immune cell clusters including: (1) Neutrophils, (2) Monocyte/Macrophages (Mono/Mac), (3) B cells, (4) Proliferating Neutrophils, (5) Proliferating Myeloid cells, (6) T/NK cells, and (7) Dendritic cells were identified and had specific changes to their populations over the injury time course (**Fig 1D–1F**). All clusters were assigned identities based on the expression of known immune markers (**Fig 1F**). Specifically, cluster 1 was labeled as neutrophils due to high expression of *S100a8* and *S100a9*. Cluster 2 was labeled as monocytes/macrophages (Mono/Mac) due to high expression levels of *Csfr1*, *Cd14* and *Adgre1*. Cluster 3 was labeled B cells for robust expression of *Ighm*, *Cd19* and *Cd79a*. Clusters 4 and 5 were high in makers of proliferation and cell cycle (*Top2a*, *Mki67*) and cytoskeleton rearrangement (*Stmn1*, *Tubb*), and were classified as proliferating neutrophils and myeloid, respectively based on their distinct expression of neutrophil markers *S100a8/9* in 4 and macrophage marker *Csfr1* in 5. T/NK cells clustered together in population 6 and were labeled based on their expression of *Nkg7* and *Thy1*. Lastly, cells in cluster 7 were classified as dendritic cells due to their high expression of *Siglech* and *Ccr9* (**Fig 1F**).

In uninjured joints, the proportion of the immune cell population represented by neutrophils was the largest in both strains, accounting for 58.7% and 71.9% of the total immune cells sequenced in B6 and MRL, respectively (**Fig 2A**). The remainder of the populations segregated as follows in uninjured B6 and MRLs: 10.1% and 9.1% Mono/Macs, 18.8% and 7.6% B cells, 3% and 3.5% proliferating myeloid, 5.3% and 3.5% proliferating neutrophils; 3% and 3.9% T/NK; ~1% and 0.5% dendritic cells. After injury, strain specific trends were observed for several cell types, with the largest population shifts occurring at D3 for nearly all identified cell types (**Fig 2A** and **S1 Table**). Prior to injury, the Mono/Mac populations were comparable in both strains, but a significant shift was observed at D3 post injury, where the proportion of these cells increased to 62.4% in MRL, and 35.5% in B6, then decreased by D7 (**Fig 2A** and **S1 Table**). At D3, the proportion of neutrophils sequenced decreased by 48.5% and 14.2% from baseline levels, in MRL and B6, respectively (**Fig 2A**). Additionally, the proportion of total proliferating myeloid cells increased by 5.2% from baseline in MRL, and 2.4% in B6 (**S1 Table**).

Since neutrophils and Mono/Macs showed the most dramatic population shifts and strain differences after injury, we focused our analysis on these two cell types. Flow cytometry confirmed that the proportion of CD45$^+$CD11b$^+$Ly6c$^+$ Mono/Macs in the synovial joint gradually increased in both strains, peaking at D3, and returning close to pre-injury levels by 4W (**Fig 2B and 2C**). A corresponding decrease in neutrophils was also observed by flow cytometry (**Fig 2B and 2D**). We also noted that MRLs had a significantly higher proportion of Mono/Macs at D1, D3 and D7 compared to B6 while B6 mice had more neutrophils than MRL specifically at D3 (**Fig 2A, 2C and 2D**).

### Neutrophils display strain specific changes in response to knee injury

Neutrophils were the major immune cell type identified in both MRL and B6 joints. These cells showed enrichment for several immune modulators including *Il1b*, a key cytokine

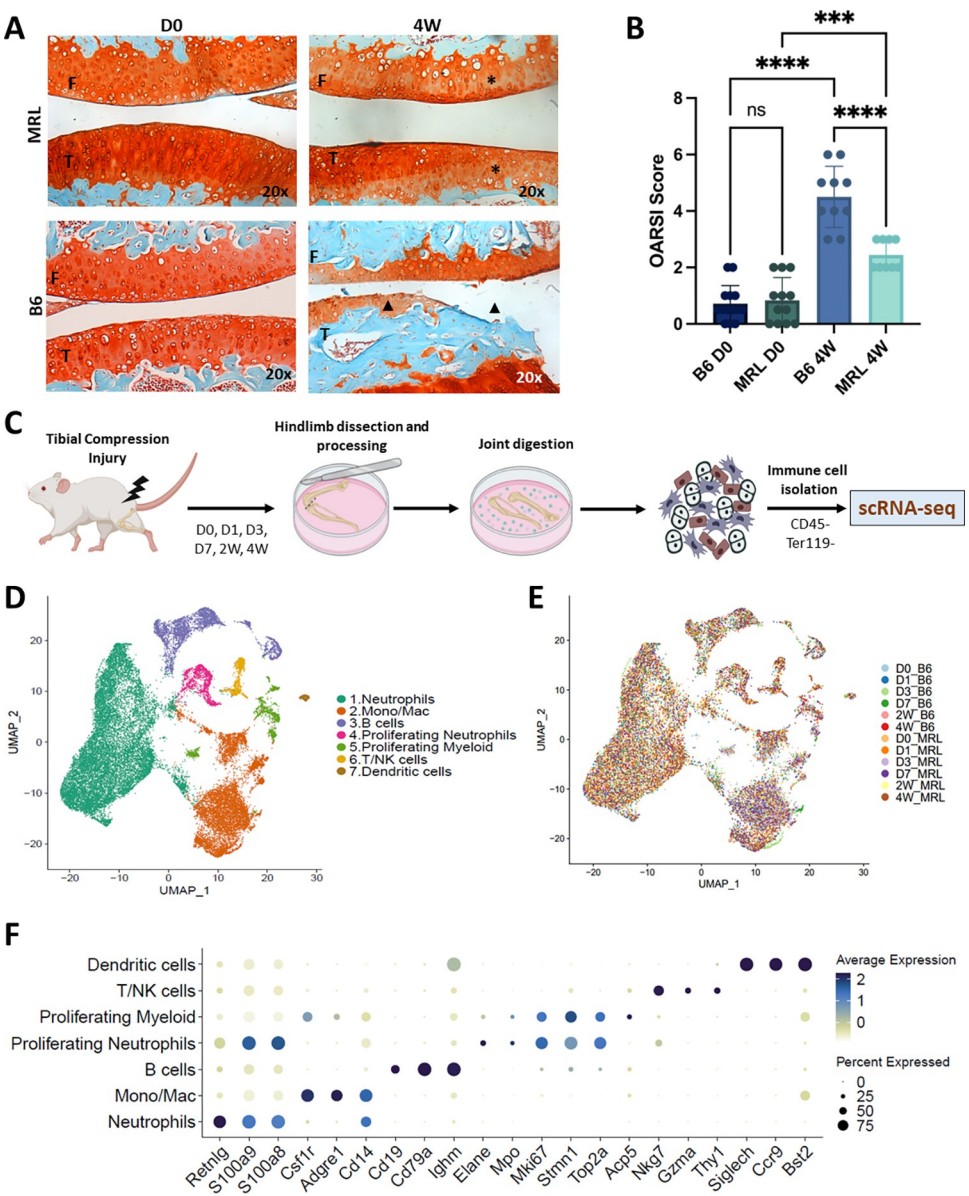

**Fig 1. Assessment of changes associated with PTOA onset in MRL and B6.** (A) PTOA resistant MRL (top row) showed little loss of staining after injury in the articular cartilage of the femur and tibia (red) indicating little to no loss of proteoglycan content in the cartilage matrix (top right, * asterisk). PTOA susceptible B6 (bottom row) showed severe degradation of bone (blue,) and cartilage (red) in the tibia, and loss of some cartilage in the femur after injury (bottom right, ▲ triangle). Scale bar = 100μm. Magnification 20X. F-Femur, T-Tibia. n = 5 / group. (B) Blinded OARSI scoring of uninjured (D0) and injured (4W) B6 and MRL joints. (*** p<0.001; ****p<0.0001). (C) Schematic of the scRNA-seq pipeline. Uninjured murine joints were collected at Day 0 (D0), injured joints were collected at Days 1- (D1), 3- (D3), 7- (D7) days, 2- (2W) and 4- (4W) weeks following tibial compression and all prepared for scRNA-seq. Digested immune cells (CD45$^+$) were enriched before conducting scRNA-seq. (D) Uniform Manifold Approximation and Projection (UMAP) plot representing seven immune cell types within the synovial joint at all timepoints examined. (E) UMAP plot from panel D colored based on experimental groups. (F) Dot plots identifying specific markers for each cell type. Size of the dot indicates cellular abundance and color indicates expression.

implicated in osteoarthritis pathogenies and *Csf1*, a key regulator of monocyte to macrophage differentiation (**Fig 2A, 2D** and **S1A Fig** in **S1 File**). To rule out the possibility that the neutrophils identified in the single cell digests were from circulation and determine whether the

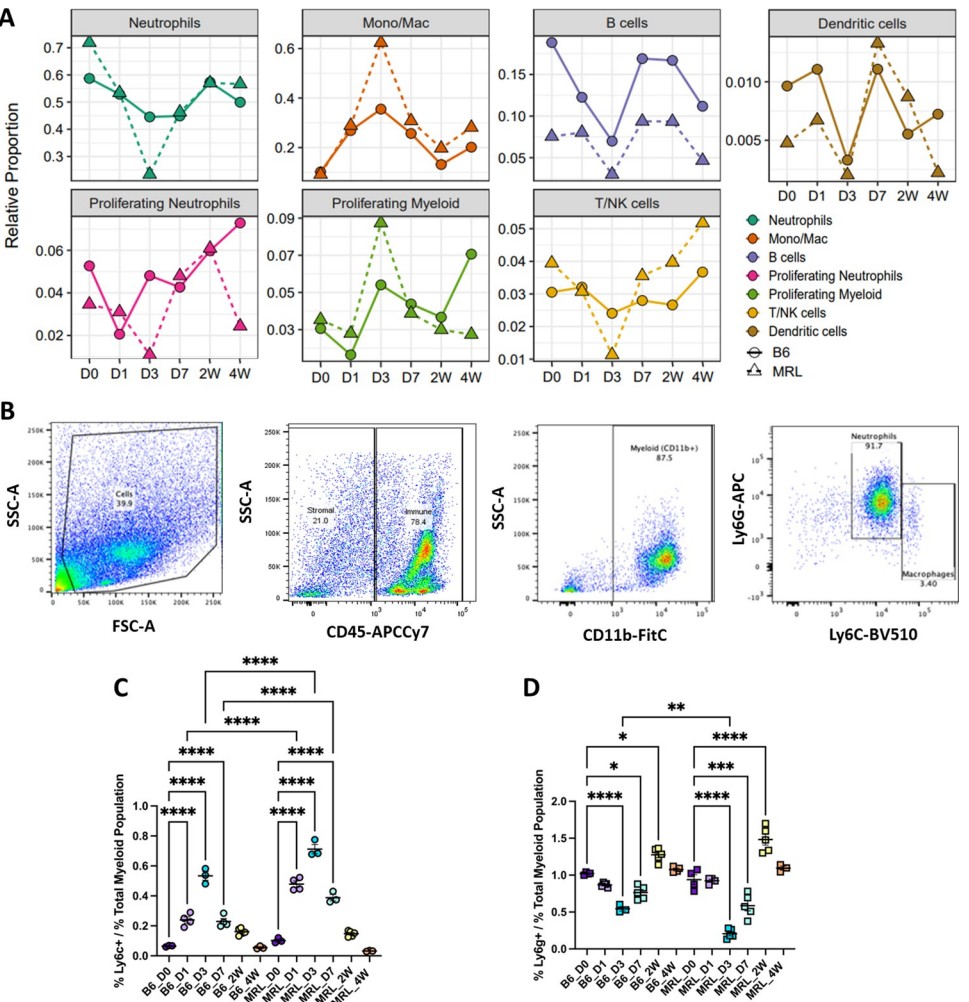

**Fig 2. Time course of immune cell population changes post injury.** (A) Percent of total for each immune population identified through scRNA-seq, determined as a proportion of all immune cells sequenced. Colors based on immune cluster identities denoted in Fig 1D. (B) Flow cytometry gating strategy for *Ly6c+* monocytes/macrophages (Mono/Mac) and neutrophil populations. (C) Trend of *Ly6c*$^+$ monocytes and macrophages after injury. (D) Trend of Ly6g$^+$ neutrophils after injury.

decrease in neutrophil proportion observed after injury corresponds to a true reduction in total neutrophils or is merely a consequence of an increase in infiltrating cells, such as Mono/Macs, flow cytometry was performed following perfusion on an additional cohort of B6. The absolute and relative number of neutrophils was analyzed at D0 and D6 post injury. We found neutrophils to contribute to ~70% of the immune cells at D0 after perfusion and a reduction in the proportion of neutrophils was observed at D6 compared to D0 *via* flow cytometric analysis (**Fig 3A**). However, our analysis showed that the total neutrophil counts were not significantly different between D0 and D6 joints (**Fig 3B**), suggesting that the reduction in the relative proportion of neutrophils after injury is likely due to infiltration of other immune cells into the joint.

Neutrophils are highly heterogeneous with several developmental stages [54]. Re-clustering of all neutrophils (cluster 1 in **Fig 1D**) identified four subtypes with distinct transcriptional profiles: (1) *Ccrl2*$^+$ neutrophils which showed strong enrichment for *Il1b*; (2) *Mmp8*$^+$

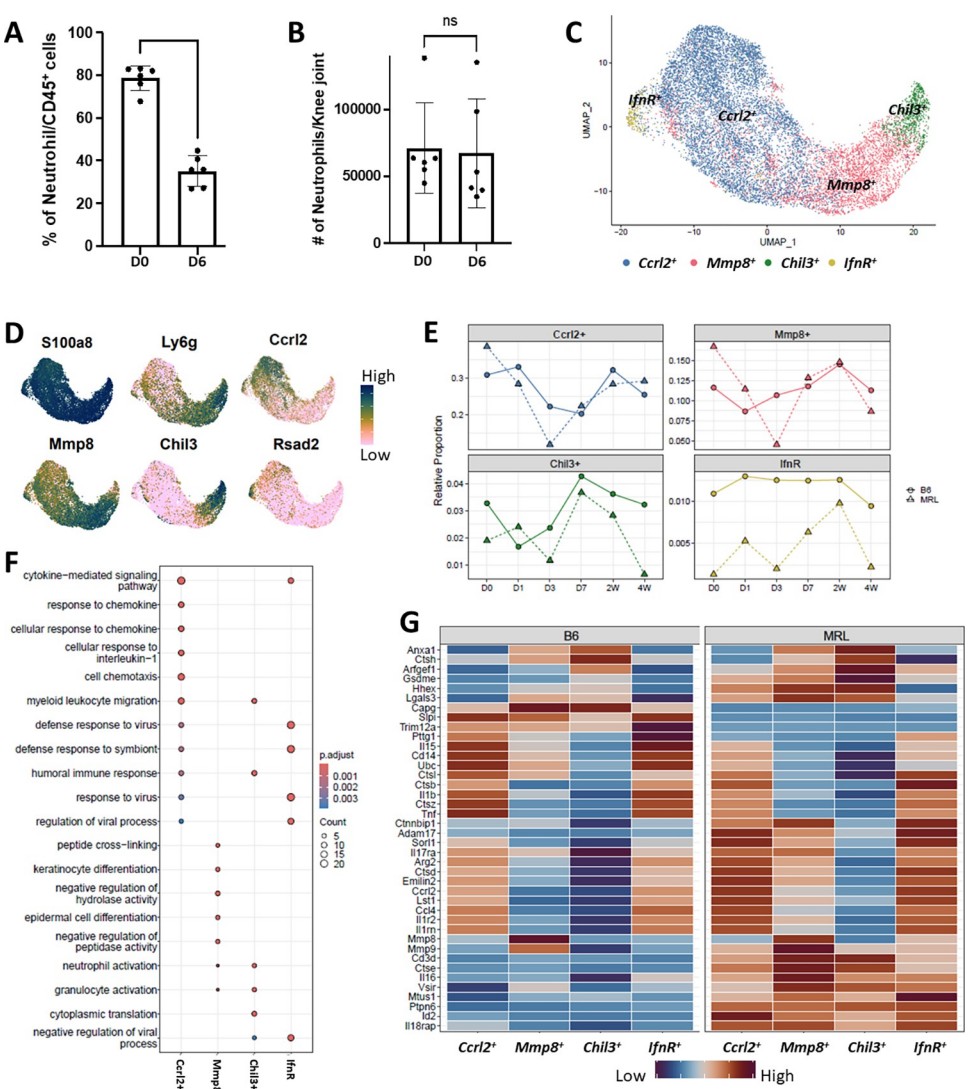

**Fig 3. Injury induced changes in MRL and B6 neutrophils.** A) Flow cytometry data showing the abundance of neutrophils relative to total immune cells in the knee joint digest after perfusion. B) Flow cytometry data showing the absolute counts of neutrophils in the knee joint digest after perfusion. C) UMAP plot showing the neutrophil subpopulations identified by scRNA-seq. D) Feature plots showing key markers of various neutrophil subpopulations. E) Changes in the proportion of various neutrophil subpopulations in response to injury in B6 and MRL, relative to total immune cells sequenced. F) Dot plot showing ontology processes enriched in each neutrophil subtype. G) Heatmap showing key genes differentially expressed between neutrophil subpopulations and between mouse strains.

neutrophils; (3) *Chil3*+ neutrophils and (4) *IfnR*+ neutrophils (**Fig 3C, 3D, S1B, S1C Fig in S1 File**). Relative proportions of all neutrophil subtypes were lower in MRL at D3 compared to BL6 (**Fig 3E**). Furthermore, gene expression signatures of *Chil3*+ neutrophils (*Chil3, Cebpe, Ngp, Ltf, Cd177*) correlated with previously established signatures of immature neutrophils while genes enriched in *Ccrl2*+ and *IfnR*+ neutrophils (*Ccl6, Csf3r, Il1b, Fth1, Ifitm1, Ifitm2, Btg1, Srgn, Msrb1*) correlated with mature neutrophils. *Mmp8*+ neutrophils had a signature (*Mmp8, Lgals3, Retnlg*) of an intermediate stage of neutrophil differentiation [55] (**S1 Fig in S1 File, S2 Table**). A gene ontology analysis identified enrichment of 'neutrophil activation' related genes in *Mmp8*+ and *Chil3*+ neutrophils while *Ccrl2*+ neutrophils showed enrichment for processes such as 'cell migration' and 'response to chemokines' (**Fig 3F**).

Further analysis of differentially expressed genes between various neutrophil subtypes showed that $Ccrl2^+$ and $IfnR^+$ neutrophils in B6 joints expressed higher levels of inflammatory cytokines, including *Il1β*, *Tnf* and *Il15*, when compared to MRL. Meanwhile, *Il18rap*, *Mmp9* and *Il1rn*, an endogenous IL1 receptor antagonist, [56] were highly expressed in MRL joints (**Fig 3G and S3 Table**). MRL joints also had increased expression of *Csf1*, a critical regulator of macrophage differentiation, when compared to B6 (**S1D Fig in S1 File**). We also observed an upregulation of genes such as *Ptgs2* and *Osm* in neutrophils after injury, in both MRL and B6 (**S1D Fig in S1 File**).

Although neutrophils constituted ~70% of immune cells at D0 in both our scRNA-seq and flow cytometry data, immunohistochemical analysis of joint tissue sections only showed a substantial number of cells expressing neutrophil markers *S100a8* or *Ly6g* in the joint *after* injury (**S2A, S2B Fig in S1 File**). To determine if the neutrophils in our digests were bone marrow-derived, we computationally compared scRNA-seq derived transcriptome profiles of the neutrophil populations from the D0 synovial joint digest to BM derived neutrophils. Similar to synovial joints, $Ccrl2^+$, $Mmp8^+$, $Chil3^+$, *IfnR* and proliferating neutrophils were also detected in BM however, the synovial joint had a significantly higher proportion of mature $Ccrl2^+$ neutrophils while the BM had more proliferating and immature neutrophils (**S2C-S2F Fig in S1 File**). We also found that synovial neutrophils expressed higher levels of inflammatory cytokines such as *Tnf*, *Il1β*, *Ccl3* and *Ccl4* when compared to BM derived neutrophils. BM derived neutrophils also showed enrichment for immature neutrophil markers *Elane*, *Mpo*, *Chil3*, *Lcn2* and *Ly6g* (**S2G Fig in S1 File**) [55], suggesting that neutrophils from the joint have a different molecular profile than the BM neutrophils. Additionally, histological analysis of the digested synovial joint indicated that the bones remained intact after digestion while the soft tissue around the knee joint was completely digested (**S2H Fig in S1 File**). This suggests that the neutrophils in our digest likely originate from tissues around the synovial joint and not from BM contamination.

Although we couldn't determine the specific location of neutrophils in the synovial joint, our data suggests that neutrophils respond to knee joint injury and B6 neutrophils expressed higher levels of inflammatory cytokines such as *Il1b* and *Tnf* compared to MRL while MRL had higher expression of *Csf1*, a cytokine involved in macrophage recruitment and activation, after injury.

## Injured MRL joints harbor significantly more macrophages than B6 injured joints

The Mono/Mac population showed the most dramatic increase in both B6 and MRL after injury. To enhance our understanding of the roles that monocytes and macrophages have in PTOA onset, Mono/Mac cells from the single cell analysis were extracted and further analyzed to investigate strain-specific changes in these cell populations, longitudinally. Eleven subpopulations (**Fig 4A and 4B**) with distinct gene expression profiles (**Fig 4C and S4 Table**) were identified; all cells shared high expression of monocyte and macrophage markers *Csf1r* and *Cd14* (**S3A Fig in S1 File**). Of these eleven subpopulations, clusters 4, 5, 7, 8 and 10 had transcriptomic profiles representative of monocytes (**Fig 4A and 4C**). Clusters 4 was identified as $Ly6c2^+$ monocytes based on the robust expression of *Ly6c2* [57] and *Plac8* [58] (**Fig 4C and 4D**). Cluster 5 expressed high levels of neutrophil markers S*100a8* and *S100a9*, in addition to *Ly6c2* and *Plac8* (**Fig 4A, 4C and S3A, S3B Fig in S1 File**) and was named $S100a8^+$ monocytes. Cluster 8 cells displayed a unique expression profile. These cells showed low-moderate expression of monocyte markers such as *Ly6c2* and *Plac8*, as well as enrichment for neutrophil markers such as *S100a8* and *S100a9* and B cell markers such as *Cd79a* and *Igkc*. It has previously

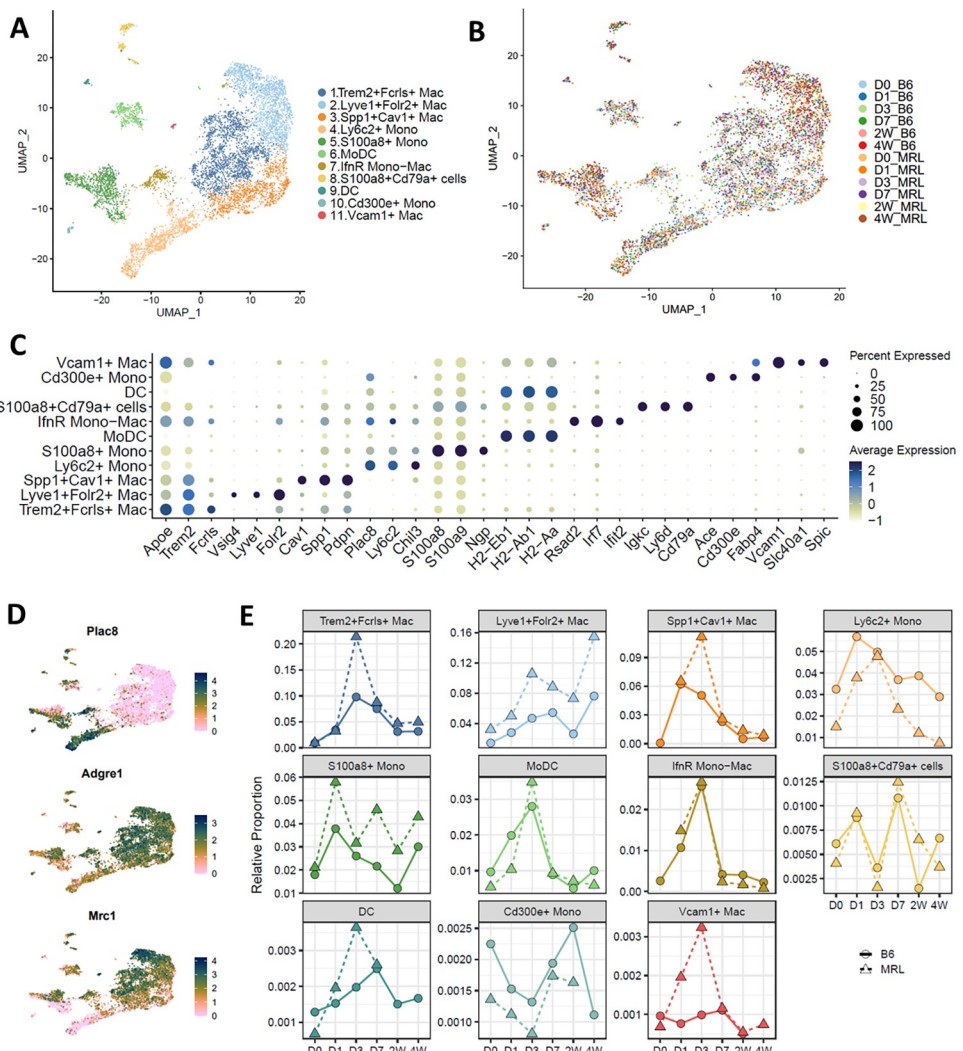

**Fig 4. Distinct gene expression profiles of monocyte and macrophage subpopulations.** (A) UMAP plot of monocyte and macrophage subpopulations identified from the parent Mono/Mac single cell cluster, colored by cell type. (B) UMAP plot of monocyte and macrophage subpopulations identified from the parent Mono/Mac single cell cluster, colored by experimental group. (C) Dot plots representing genes used to distinguish monocyte and macrophage subpopulations. (D) Feature plots of key monocyte and macrophage markers. (E) Changes of monocyte and macrophage subpopulations after injury from scRNA-seq data of B6 (solid line) and MRL (dashed line). The cell type proportions were calculated relative to all immune cells sequenced. Colors representative of clusters in Panel A.

been reported that pre/pro-B cells can differentiate into macrophages [59] and this cluster was identified as *S100a8⁺Cd79a⁺*. Cluster 7 expressed both monocyte and macrophage markers as well as high levels of genes involved in interferon signaling (**Fig 4C, 4D and S1A, S1B Fig in S1 File**). This cluster was identified as interferon responsive Mono-Mac (*IfnR* Mono-Mac). Cluster 10 showed enrichment for genes such as *Cd300e*, *Ace* and *Fabp4*, in addition to moderate *Plac8* expression and was identified as *Cd300e⁺* (**Fig 4C and S3C Fig in S1 File**). When examining changes in various monocyte populations in response to injury, we observed a sharp increase in *Ly6c2⁺* and *S100a8⁺* monocytes immediately after injury, in both strains (**Fig 4D and S5 Table**). Interestingly, B6 had a higher proportion of *Ly6c2⁺* monocytes relative to MRL at most timepoints examined while MRL had higher proportion of *S100a8⁺* monocytes (**Fig 4D and S5 Table**).

Cluster 6, monocyte-derived Dendritic Cells (MoDCs), were high in *Cd209a*, *Cd14* as well as MHC class II genes involved in antigen presentation such as *Cd74*, *H2-Ab1* and *H2-Aa* (**Fig 4C and S3A, S3B Fig in S1 File**). We also identified another small dendritic cell cluster (DC; cluster 9) which had high expression of MHC class II genes including *Cd74* and *H2-Ab1* and accounted for less than 2% of Mono/Macs at any timepoint examined (**Fig 4C and S3A-S3C Fig in S1 File**).

Clusters 1 and 3 expressed high levels of macrophage marker *Adgre1 (F4/80)* and the proportions of both clusters increased dramatically after injury in both strains (**Fig 4C–4E**). Moreover, both subpopulations had high expression of *Trem2*, a gene that has been previously shown to promote myeloid cell phagocytosis [60–62], but *Trem2* was significantly enriched in cluster 1 (**Fig 4C and S3A, S3B Fig in S1 File**). Cluster 1 also showed enrichment for *Fcrls*, *Gas6*, *Apoe*, *C1qa* and *C1qb* and was named *Trem2*+*Fcrls*+. Cluster 3 highly expressed *Cav1* (Caveolin-1) (**Fig 4C**), a gene shown to promote monocyte to macrophage differentiation [63] as well as *Spp1*, *Vim*, *Arg1*, and *S100a4;* these macrophages were labeled *Spp1*+*Cav1*+. We also noted that clusters 1 and 2 were moderately comparable transcriptionally (**S3B Fig in S1 File**) and clustered closely together in UMAP projections (**Fig 4A**). *Mrc1* (Cd206), a marker of alternatively activated macrophages, also known as M2 macrophages, was highly expressed in the *Trem2*+*Fcrls*+ cells, while the *Spp1*+*Cav1*+ population had very low expression (**Fig 4D and S3B Fig in S1 File**). The proportion of *Spp1*+*Cav1*+ population peaked at D1 in B6 while both *Trem2*+*Fcrls*+and *Spp1*+*Cav1* macrophage populations peaked at D3 in MRL (**Fig 4E and S5 Table**). Together these two macrophage subpopulations accounted for over 45% of all Mono/Macs at D3 in both strains (**S3C Fig in S1 File**).

Cluster 2 had high expression of *Mrc1*, as well as several tissue resident macrophage markers including *Lyve1*, *Folr2*, *Vsig4* and *Timd4* [64, 65]. This cluster was identified as *Lyve1*+-*Folr2*+ macrophages. *Trem2* and its ligand *Apoe* were robustly expressed in this cluster (**Fig 4C and S3A, S3B Fig in S1 File**). MRLs had a significantly higher proportion of *Lyve1*+*Folr2*+ macrophages compared to B6 at nearly all timepoints examined (**Fig 4E**). Previously, our group identified that resident *Lyve1*high macrophages localize primarily at the synovial lining within the uninjured knee joints of B6 mice in a highly organized fashion but infiltrate the synovium following injury and appear disorganized in the tissue (**S4 Fig in S1 File**) [32]. In this study, we also identified *Lyve1*+ cells at the synovial lining of the MRL joint (**S4 Fig in S1 File**) but unlike B6, these cells remained highly organized post injury. *Lyve1*+*Folr2*+ macrophages expressed high levels of bone and cartilage anabolic growth factors, including *Bmp2* and *Igf1* (**S3B Fig in S1 File**), suggesting that they may have a protective role in PTOA onset [31, 62, 66, 67] and may contribute to the resistance to PTOA seen in MRL joints.

## Trem2+ macrophages have decreased inflammatory signaling in MRLs

While *Trem2*+ *Fcrls*+ and *Spp1*+*Cav1*+ subpopulations were essentially nonexistent in the uninjured joint, these populations displayed the largest shifts post injury in both strains (**Fig 4D**) suggesting that they are recruited to the joint tissue after injury. We observed *Ly6c2* expression in *Trem2*+ *Fcrls*+ and *Spp1*+*Cav1*+ macrophages at D1 (**S5A Fig in S1 File**) as well as an increase in *Mrc1* expression primarily in *Trem2*+ *Fcrls*+ cells at D3-D7 suggesting that they are monocyte-derived, M2 polarized macrophages (**Fig 5A and S5B Fig in S1 File**). This observation matched the population shifts observed in the *Ly6c*+ Mono and *Trem2*+ *Fcrls*+ subpopulations (**Fig 4E**). However, it has been suggested that resident synovial macrophages may also polarize into *Trem2*+ *Fcrls*+ and *Spp1*+*Cav1*+ macrophage subpopulations [68]. To determine the differentiation trajectory of these recruited *Trem2*+ macrophage subpopulations, we conducted a pseudo-time trajectory analysis with *Trem2*+ *Fcrls*+, *Spp1*+*Cav1*+, *IfnR*+ and *Lyve1*+

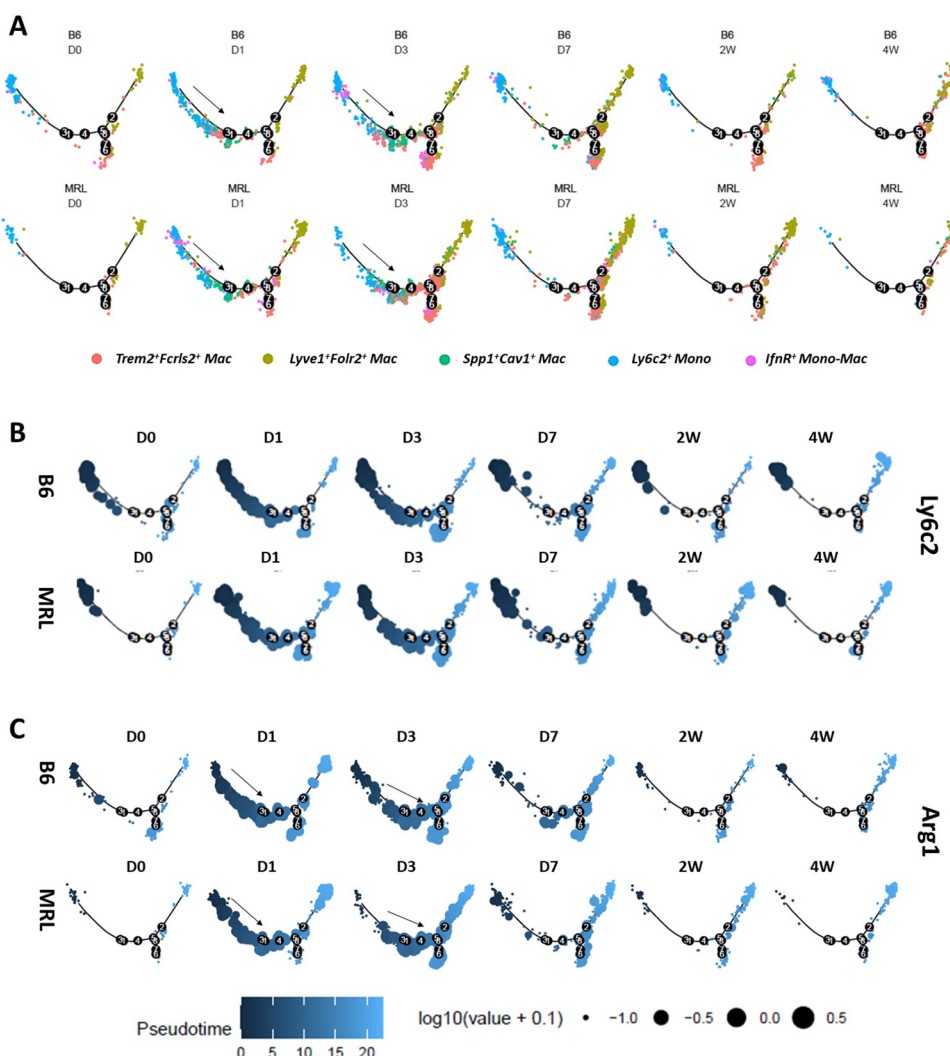

**Fig 5. Pseudo-time differentiation trajectory analysis of *Trem2⁺Fcrls⁺* and *Spp1⁺Cav1⁺* macrophages from MRL and B6.** (A) Pseudo-time trajectory analysis was conducted to determine potential origin of *Trem2⁺Fcrls⁺* and *Spp1⁺Cav1⁺* macrophages from *Ly6c2⁺* monocytes or tissue resident macrophages. The relative position of cells across the pseudo-time differentiation trajectory is depicted in the figure. Each point is a cell and is colored according to its cluster identity. For both MRL and B6, cells along the trajectory were divided into six groupings based on experimental timepoints (D0-4W). An expansion of *Ly6c2⁺* monocytes along the trajectory towards macrophages was observed after injury, primarily at D1 and D3 in both strains (indicated by arrows). B) Superimposition of the expression of monocyte marker *Ly6c2* on the pseudo-time trajectory. Each point is a cell and is colored according to its pseudo-time value. Circle size represents the gene expression level. C) Superimposition of the expression of *Arg1*, a gene specifically enriched in *Spp1⁺Cav1⁺* macrophages at D1-D3, on the pseudo-time trajectory. Expansion of cell populations expressing high levels of *Arg1* in the monocyte to macrophage direction was observed after injury (indicated by arrows).

macrophages and *Ly6c2⁺* monocytes; *S100a8⁺* monocytes were excluded from this analysis as they appeared to be highly distant from *Trem2⁺* macrophages [32]. Pseudo-time analysis showed an expansion of *Ly6c2⁺* monocytes along the differentiation trajectory in the direction of *Trem2⁺ Fcrls⁺* and *Spp1⁺Cav1⁺* macrophages, primarily at D1 and D3 (**Fig 5A and S6 Fig in S1 File**). Also, *Ly6c2* expression in these expanding cells coincided with the expression of *Trem2*, *Arg1*, a gene enriched in *Spp1⁺Cav1⁺* specifically at D1 and D3, and macrophage marker *Adgre1* (F4/80) (**Fig 5 and S6 Fig in S1 File**). We observed an increase in *Mrc1*

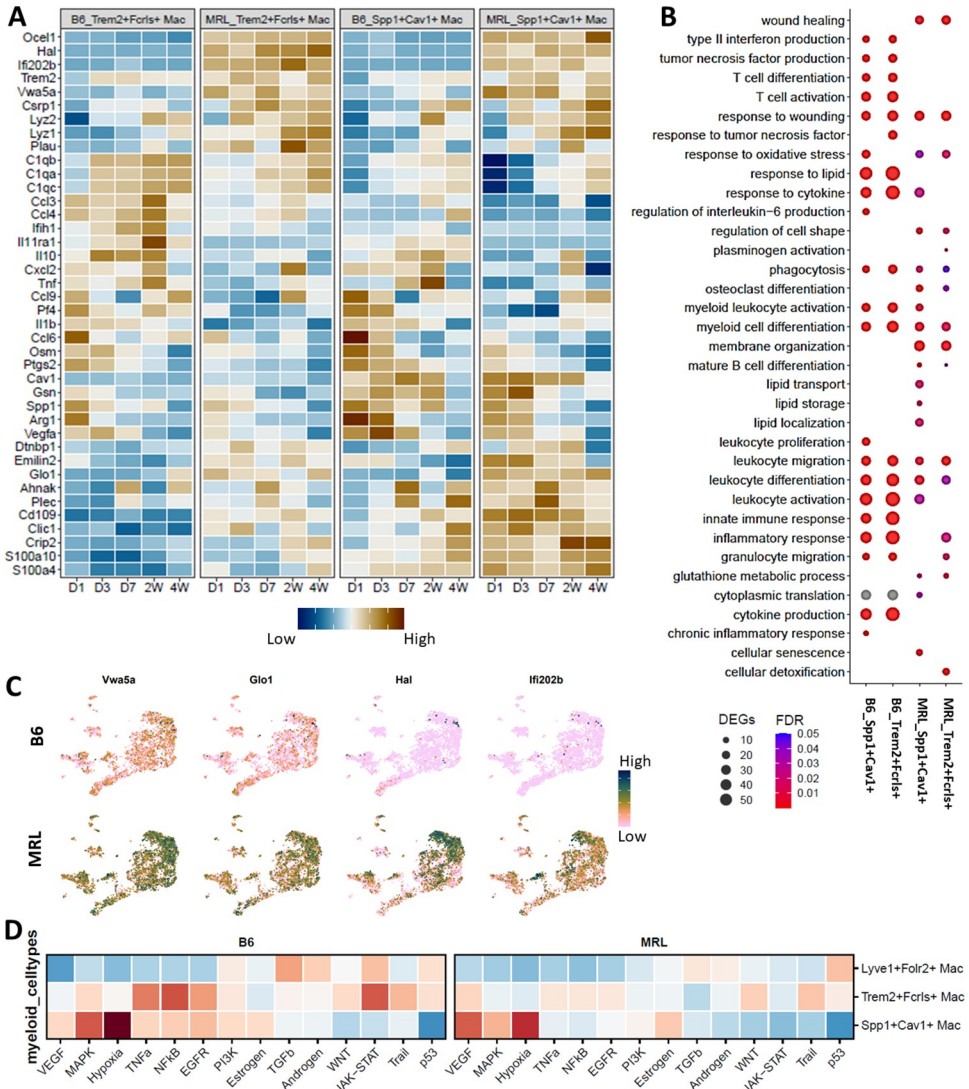

**Fig 6. Trem2 expressing macrophages have distinct transcriptional profiles and activation states in MRL and B6.**
(A) Heatmap of a subset of genes differentially expressed in *Trem2⁺ Fcrls⁺* and *Spp1⁺Cav1⁺* macrophage populations from MRL and B6 joints (B) Enriched ontology terms associated with genes differentially expressed between MRL and B6 in *Trem2⁺ Fcrls⁺* and *Spp1⁺Cav1⁺* clusters. (C) Selected genes that are highly or exclusively expressed in MRL *Trem2⁺ Fcrls⁺ and Spp1⁺ Cav1⁺* populations. (D) Pathways enriched in Trem2-expressing macrophage subpopulations.

expression in these expanding cells primarily at D3, suggesting a monocytic origin for *Trem2⁺ Fcrls⁺* and *Spp1⁺Cav1⁺* macrophages (**S6 Fig in S1 File**). However, *Lyve1⁺* resident macrophages may also polarize into *Trem2⁺Fcrls⁺* and *Spp1⁺Cav1⁺* phenotype especially at later post-injury timepoints (**Fig 5 and S6 Fig in S1 File**).

To better understand transcriptomic changes in *Trem2⁺Fcrls⁺* and *Spp1⁺Cav1⁺* macrophages between strains and in response to injury, we performed differential expression analysis using Seurat (**S6 Table**). Our analysis showed that *Trem2⁺Fcrls⁺* and *Spp1⁺Cav1⁺* macrophage subpopulations from B6 expressed significantly higher levels of genes associated with cytokine and pro-inflammatory signaling including *Ccl3*, *Ccl4*, *Ccl6*, *Ccl9*, *Il1b*, *Osm* and *Tnf* relative to MRL (**Fig 6A**). Gene ontology analysis of genes upregulated in B6 *Trem2⁺Fcrls⁺* compared to

MRL identified enrichment for biological processes such as 'leukocyte migration', 'cytokine production', 'inflammatory response', 'type II interferon production' and 'tumor necrosis factor production' (Fig 6B), indicating an enrichment of pro-inflammatory functions. These processes were also enriched in B6 *Spp1*+*Cav1*+ macrophages compared to MRL. In addition, *Spp1*+*Cav1*+ macrophages from B6 showed enrichment for genes associated with 'chronic inflammatory response', 'leukocyte proliferation' and 'regulation of interleukin-6 production' compared to MRL (Fig 6B).

Genes upregulated in MRL *Trem2*+*Fcrls*+ and *Spp1*+*Cav1*+ macrophages compared to B6 showed enrichment for processes such as 'wound healing', 'leukocyte activation', 'osteoclast differentiation', 'phagocytosis', 'mature B cell differentiation', 'response to oxidative stress', 'regulation of cell shape', 'membrane organization' and 'glutathione metabolic process' (Fig 6B). In addition, *Spp1*+*Cav1*+ macrophages from MRL showed enrichment for several lipid-associated processes including 'lipid transport', 'lipid storage' and 'lipid localization' (Fig 6B). Lipid transport or metabolism-associated genes enriched in MRL included *Cav1*, *Pltp*, *Trem2*, *Abcg1*, *Plin2*, *Aig1* and *Vps13c* (S6 Table). We also identified multiple genes that were significantly higher (*Vwa5a* and *Glo1*) or exclusively (*Hal*, and *Ifi202b*) expressed in MRLs (Fig 6A and 6C). Many of these genes appeared to be differentially expressed between all MRL and B6 myeloid subpopulations and these genes may represent inherent strain specific differences (Fig 6A, 6C and S7A Fig in S1 File).

A pathway enrichment analysis revealed differential enrichment for several pathways between macrophage subpopulations and mouse strains (Fig 6D). *Trem2*+*Fcrls*+ macrophages from B6 showed strong enrichment for TNF, NFKB, JAK/STAT and EGFR signaling compared to MRL, while MRL macrophages showed enrichment for VEGF signaling (Fig 6D). B6 *Spp1*+*Cav1*+ macrophages showed enrichment for hypoxia pathway, MAPK, TNF, NFKB, EGFR, PI3K and JAK/STAT signaling compared to MRL (Fig 6D). We also observed that, in both strains, TNF, NFKB, JAK/STAT and EGFR signaling was enriched in *Trem2*+ *Fcrls*+ macrophages relative to *Spp1*+*Cav1*+ macrophages while *Spp1*+*Cav1*+ macrophages showed enrichment for hypoxia, MAPK and VEGF signaling (Fig 6D). Consistent with this, a transcription factor binding motif analysis identified hypoxia transcription factors *Hif1a* and *Hif2a* (Epas1) as enriched in *Spp1*+*Cav1*+ macrophages, with the highest enrichment in B6 mice (S7B Fig in S1 File). *Hif1α* gene expression was also enriched in *Spp1*+*Cav1*+ macrophages along with multiple genes encoding for glycolytic enzymes such as *Ldha* and *Eno1*, potentially regulated by Hif1α [69, 70] (S7C Fig in S1 File). In addition to *Mrc1*, *Trem2*+*Fcrls*+ macrophages showed strong enrichment for several other genes highly expressed in M2 macrophages such as *Il10*, *Marcks* and *Cd83*, which were also shared by *Lyve1*+*Folr2*+ resident macrophages (S3B, S7C Figs in S1 File).

## Sustained *Trem2*+ M2 macrophage populations in MRLs may promote tissue repair

To further identify trends in macrophage infiltration and M2 activation that may contribute to MRL's resilience to PTOA development, *Mrc1*- and *Trem2*– expressing macrophage subpopulations from scRNA-seq data were compared between B6 and MRL across all timepoints (Fig 7A). Uninjured MRL joints had a higher proportion of *Mrc1*+ cells than B6 and sustained a consistently higher proportion at all timepoints indicating an increase in macrophage polarize towards an M2 phenotype in this strain (Fig 7A). The majority of *Mrc1*+ cells also expressed high levels of *Trem2* in both strains (Fig 7A, S3B Fig in S1 File). The increased levels of *Mrc1*+*Trem2*+ cells in the MRL joints suggest that M2 macrophages expressing *Trem2* may play a vital role in injury response and promote the enhanced healing associated with this strain.

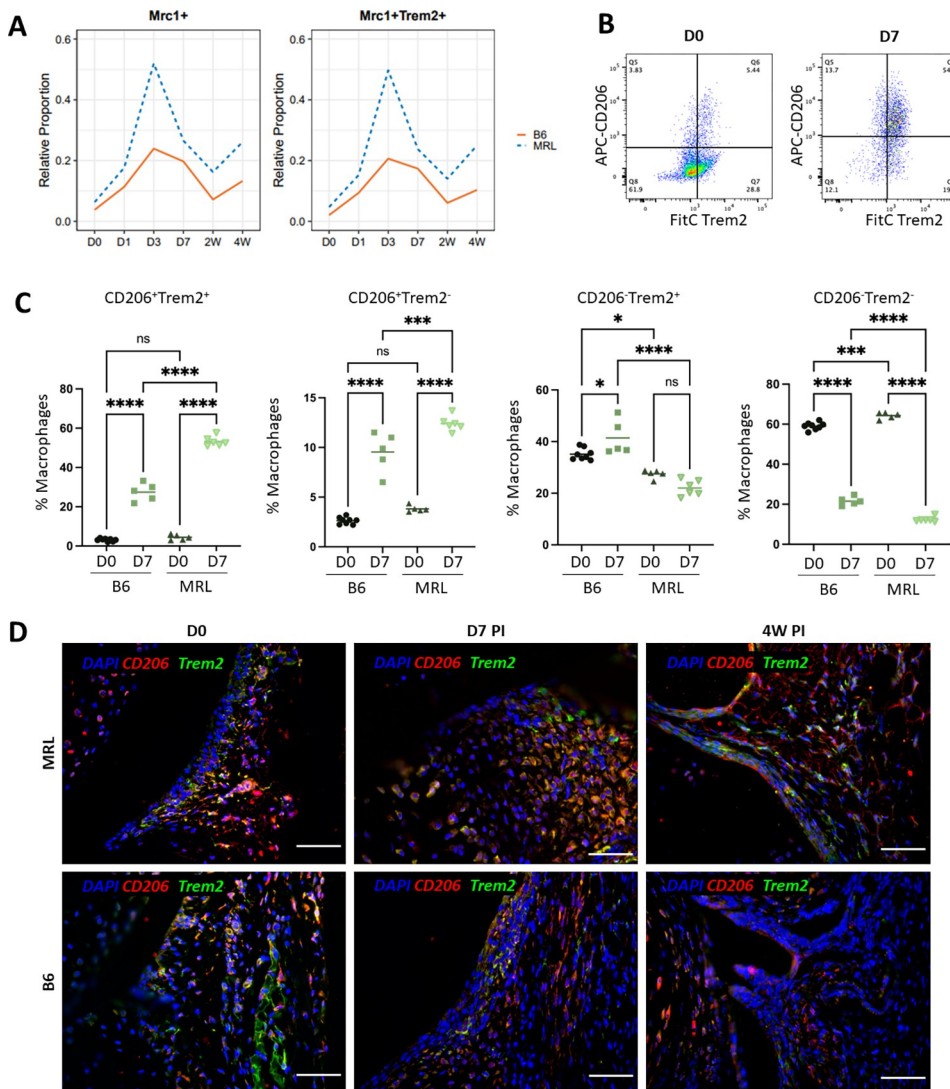

**Fig 7. Flow cytometry and immunohistochemistry analysis confirm an increased presence of CD206⁺Trem2⁺ macrophages in MRL knee joints.** A) Proportion of *Mrc1⁺ and Mrc1⁺Trem2⁺* Mono/Macs relative to total immune cells sequenced in both B6 (solid red line) and MRL (dotted blue line) across all injury timepoints. (B) Representative gating scheme for the analysis of macrophages (CD45⁺F4/80⁺) that have shifts in CD206 and Trem2 expression between strains within digested synovial joint immune populations at D7. (C) Proportion of cells in B6 and MRL with macrophage expression profiles of CD45⁺F4/80⁺CD206⁺Trem2⁺, n = 5; **p<0.01, ***p<0.001, ****p<0.0001, n = 3–5 / group. (D) Immunohistochemistry evaluation of macrophages expressing Trem2 in MRL and B6 at D0 (uninjured), D7 post injury and 4W post injury. n = 5 / group. Scale Bars = 200μm, 20x Magnification, Red–CD206, Green–Trem2, Blue–Nuclei.

Further validation of *Mrc1* (CD206) and *Trem2* expression was conducted by flow cytometry (**Fig 7B and 7C**). Viable CD45⁺F4/80⁺ cells were gated as the macrophage population and then analyzed for shifts in CD206 (Mrc1) and Trem2 protein expression in uninjured and D7 post injury joints (**Fig 7B and 7C**). Consistent with the scRNA-seq data, we observed a large spike in the CD206⁺Trem2⁺ population at D7 in both strains. In addition, MRLs had significantly more CD206⁺Trem2⁺ macrophages than B6 at D7 post injury (**Fig 7C**). Shifts in CD206⁺Trem2⁺ macrophages were also validated at the protein level through immunofluorescence of synovial joints (**Fig 7D**). In uninjured joints, MRL had a stronger Trem2 expression

than B6 (**Fig 7D**; **D0**), and robust expression of CD206 was seen throughout the synovium of the knee joint compared to B6. At D7, CD206$^+$Trem2$^+$ expression was higher in MRL joints than B6 as indicated by the yellow co-expression of Trem2 and CD206 (**Fig 7D**).

## Discussion

This study sheds new light on the important modulatory role immune cells have in the prevention or onset of chronic joint degeneration. scRNA-seq and a non-surgical injury method that replicates ACL injury in humans allowed the unbiased examination of the immune heterogeneity in the synovial knee joint of PTOA-susceptible B6 mice and PTOA-resistant MRL mice [32, 35, 71, 72]. Previous studies have implicated infiltrating myeloid-derived populations, such as neutrophils and monocytes, as culprits of a pro-inflammatory joint state during osteoarthritis progression [3, 73, 74]. These cells are responsible for the production of inflammatory cytokines and chemokines, such as *IL-1β*, *TNFα*, *IL-6*, *IL-10*, and *IL-15*, as well as many others from the CCL/CXCL family [74]. Many of these molecular signals may be responsible for the infiltration of innate (macrophages, neutrophils, NK) and adaptive (T, B) immune cells into the synovial joint. Here, we characterized resident and infiltrating monocyte and macrophage subpopulations as well as neutrophils present in the synovial knee joint of MRL and B6 mice. Previously, *Ly6c$^{high}$* monocytes have been shown to be recruited to the joint in response to traumatic knee injury and act as pro-inflammatory effector cells in tissues with perturbed homeostasis [75]. We found B6 joints to consistently have higher numbers of *Ly6c$^+$* monocytes than MRL while the MRLs had an increased number of M2 macrophages. Increased presence of *Ly6c$^+$* monocytes in B6 may have contributed to the development of a pro-inflammatory microenvironment in these mice, interfering with their ability to repair damaged tissue.

A major M2-like macrophage population identified in the synovial joint was the resident *Lyve1$^+$* macrophages. In addition to established tissue resident macrophage markers such as *Lyve1*, *Folr2* and *Vsig4*, *Lyve1+* macrophages expressed *Trem2*, its ligand *Apoe* and several growth factors with potential chondroprotective functions such as *Igf1* and *Bmp2* [76, 77]. *Trem2* expression has previously been associated with macrophages responsible for forming a protective barrier in synovial joints [25, 32]. *Trem2$^+$* alternatively activated macrophages have been shown to drive an anti-inflammatory tissue environment and to promote damage repair *via* stromal cell interactions in rheumatoid arthritic (RA) joints as well as in other tissues [78, 79], therefore the increase in the *Trem2$^+$* M2-like macrophage population after an injury is likely to confer a protective phenotype. Although the proportion of *Lyve1$^+$* macrophages did not change considerably over time, MRL constantly had more *Lyve1$^+$* macrophages than B6 at all timepoints. We also found that *Lyve1$^+$* macrophages from B6 expressed higher levels of inflammatory cytokines (*Tnf*, *Ccl3*, *Ccl4*) than MRL, suggesting that *Lyve1$^+$* macrophages in B6 are likely proinflammatory.

In addition to resident *Lyve1$^+$* macrophages, *Trem2* was also expressed in *Trem2$^+$Fcrls$^+$* and *Spp1$^+$Cav1$^+$* macrophage clusters, with significantly higher expression in *Trem2$^+$Fcrls$^+$* cluster than *Spp1$^+$Cav1$^+$* cluster. Starting at D1 post injury, MRL and B6 synovium experienced an increase in the proportion of both *Trem2$^+$Fcrls$^+$* and *Spp1$^+$Cav1$^+$* macrophages. Both these clusters expressed low levels of *Ly6c2* at D1 indicating these cells were monocyte derived and emerged into the synovial joints after injury. Further gene and ontology enrichment analysis of *Trem2$^+$* recruited macrophages identified a more pro-inflammatory molecular phenotype in B6 cells. Specifically, *Trem2$^+$* infiltrating macrophages in B6 were highly associated with proinflammatory cytokine expression and ontologies associated with inflammatory response. In contrast, several genes highly expressed in MRLs were associated with biological processes

such as wound healing and response to oxidative stress, suggesting that these macrophages may contribute to the enhanced healing phenotype observed in MRL.

*Trem2*⁺*Fcrls*⁺ macrophages also expressed high levels of M2 marker *Mrc1* (CD206) and several other genes enriched in M2 macrophages including *Cd83*, *Marcks* and *Apoe* suggesting that this population shares some similar functions with the *Lyve1*⁺ population. Flow cytometry analysis confirmed that MRL has significantly more CD206⁺*Trem2*⁺macrophages than B6 at D7. The sustained level of *Trem2*⁺ macrophages in MRLs suggests that MRLs are better equipped to respond to injury through recruitment of hematopoietic progenitors and M2 polarization *via* cytokine signaling, such that phagocytosis of apoptotic cells induced by initial joint inflammation is more effective and promotes healing [80, 81]. Our study suggests that activation of Trem2⁺ macrophages with a pro-healing profile may help with knee joint repair in PTOA patients as observed in RA patients [62].

*Spp1*⁺*Cav1*⁺ macrophages shared molecular signatures (*Spp1*, *Fn1*, *Arg1*, *Capg* etc.) with previously described *Spp1*⁺ pro-fibrotic macrophages [82]. In line with the findings by Hoeft *et al*, *Spp1*⁺ macrophages showed strong enrichment for Hypoxia-inducible factor 1α (*Hif1α*) signaling [82]. *Hif1α* promotes the switch from oxidative phosphorylation to glycolysis so that cells can continue to produce ATP when oxygen is limited, as oxygen is not required for glycolysis [83]. Consistent with *Hif1α* activation, we observed an increase in the expression of glycolytic enzymes including *Ldha*, *Eno1* and *Aldoa* in *Spp1*⁺*Cav1*⁺ macrophages, all of which had higher expression in MRL compared to B6. Further studies are required to understand the role of *Spp1*⁺ macrophages in injured joint and to determine if increased expression of these glycolytic enzymes in *Spp1*⁺ macrophages help with the enhanced healing or PTOA resistance observed in MRL. Knight *et al* suggested that these pro-fibrotic macrophages arise from synovial resident macrophages after injury [68]. However, our monocle trajectory analysis suggested that *Ly6c2*⁺ monocytes could also differentiate into *Spp1*⁺ macrophages especially at early post-injury timepoints, which is consistent with the findings by Ramachandran *et al*, in liver cirrhosis [84]. Thorough *in vivo* fate mapping studies are required to elucidate the true origins of these cells.

We also identified several genes consistently upregulated in all monocytes and macrophages from MRL, many of which had the highest expression in *Trem2*⁺ macrophages. Specifically, *Glo1* has been shown to help inhibit inflammation by managing methylglyoxal levels produced by macrophages, thus inhibiting cell death and cytokine production [85]. An increase in *Glo1* expression in *Trem2*⁺ macrophages from MRL indicates that these cells may play an essential role in dampening proinflammatory signaling in MRL most likely through reactive oxygen compounds or metabolites by locally damaged cells. In addition, we identified *Vwa5a* as a gene highly enriched in MRL macrophages compared to B6. *Vwa5a* has been shown to be associated with longevity [86, 87] and tumor suppression [88]. Increased expression of these genes may also contribute to the PTOA-resistant phenotype observed in MRL. These molecules can be taken advantage of therapeutically upon further characterization of their function in the context of PTOA. Neutrophils were a major immune population identified in our data. Although we failed to detect a considerable number of neutrophils in the knee joint tissues *via* IHC, we were able to rule out contamination from the circulation or the BM as possible origins of these neutrophils. This suggests that these neutrophils likely reside within the joint or adjacent tissues such as fat pad or the bone. Neutrophils expressed high levels of inflammatory cytokines (*Il1b*, *Tnf*, *Osm* etc.) and matrix degrading enzymes (*Mmp8*, *Mmp9* etc.) in both MRL and B6. We also noted that MRL neutrophils had lower expression of *Il1β* compared to B6 but, had higher expression of endogenous Il1 receptor antagonist *Il1rn*. In addition, MRL neutrophils expressed higher levels of *Csf1*, a key growth factor required for macrophage differentiation [89], than B6, which may have contributed to the increased

presence of macrophages in MRL joints. This indicates that the presence of neutrophils in the joint may also contribute to differences in injury outcomes observed in MRL and B6. Our data highlights gene expression changes in response to injury and strain specific differences in neutrophils. However, further studies are needed to localize neutrophils in the articular joint forming tissues and understand their specific role in PTOA pathogenesis.

This study has several limitations. While the insight from this study identifies specific cellular responses to PTOA in mice and several potential therapeutic targets, there are slight differences in the immune system and cartilage/bone physiology between human and mouse [90] that needs to be taken into account while developing therapies against these targets. Single cell RNA sequencing is an invaluable tool for characterizing cellular populations and determining cell type-specific transcriptomic changes in exploratory research studies. While this technology has many perks including identifying unique and rare subsets of cells, the true cellular function and environmental interactions of the immune system and stromal cells needs further characterization. Furthermore, we currently can't distinguish between BM/circulating neutrophils from neutrophils that have infiltrated the joint tissue, but future lineage tracing experiments will aid in this endeavor. Thus, further exploration of the Trem2[+] macrophages and genes enriched in MRL in both clinical samples and other pre-clinical models are essential in helping to determine how some individuals fully recover from an ACL injury without developing PTOA, while others succumb to this degenerative disorder and also to determine the therapeutic potential of targeting these cell types and genes. Also, it is likely that we missed some immune cell types in this study such as fully mature granulocytes due to the limitations of the sequencing technology we used. However, the single cell data generated in this study, describing temporal changes in immune cells from MRL and B6 in response to knee injury, would add great value to the research community and could be hypothesis generating for researchers working in this area.

## Conclusions

This study represents the first report describing fundamental molecular and cellular differences in neutrophil and macrophage subpopulations and macrophage polarization in the injured joint that may set the super-healer MRL strain apart from B6. Here we identified *Trem2*[+] macrophage populations as cells with a potential role in tissue repair and/or PTOA resistance in MRL. Consistent with previous studies [15], our study showed that MRL has reduced expression of inflammatory genes and identified several inflammatory genes upregulated in B6 compared to MRL, which may have played a role in joint degeneration observed in B6 after injury. We also identified several genes including *Glo1* and *Vwa5a* with increased expression in MRL macrophages.

## Supporting information

**S1 File.**
(PDF)

**S1 Table. Proportion of various immune cells in B6 and MRL knee joints at D0, D1, D3, D7, 2W and 4W.**
(XLSX)

**S2 Table. Genes enriched in each neutrophil subtypes relative to others.**
(XLSX)

**S3 Table. Genes differentially expressed between MRL and B6 neutrophil subtypes.**
(XLSX)

**S4 Table. Genes enriched in each macrophage cluster relative to others.**
(XLSX)

**S5 Table. Proportion of various mono/mac subpopulations in B6 and MRL relative to total immune cells sequenced.**
(XLSX)

**S6 Table. Genes differntially expressed between various macrophage subpopulations from MRL and B6 mice.**
(XLSX)

## Acknowledgments

This work was performed under the auspices of the U.S. Department of Energy by Lawrence Livermore National Laboratory under Contract DE-AC52-07NA27344.

## Author Contributions

**Conceptualization:** Jillian L. McCool, Aimy Sebastian, Blaine A. Christiansen, Gabriela G. Loots.

**Data curation:** Aimy Sebastian, Nicholas R. Hum.

**Formal analysis:** Jillian L. McCool, Aimy Sebastian.

**Funding acquisition:** Aimy Sebastian, Gabriela G. Loots.

**Investigation:** Jillian L. McCool, Aimy Sebastian, Oscar A. Davalos, Deepa K. Murugesh.

**Methodology:** Jillian L. McCool, Aimy Sebastian, Nicholas R. Hum, Stephen P. Wilson, Oscar A. Davalos, Deepa K. Murugesh, Beheshta Amiri, Cesar Morfin, Blaine A. Christiansen.

**Project administration:** Gabriela G. Loots.

**Resources:** Gabriela G. Loots.

**Supervision:** Aimy Sebastian, Blaine A. Christiansen, Gabriela G. Loots.

**Validation:** Aimy Sebastian.

**Visualization:** Aimy Sebastian.

**Writing – original draft:** Jillian L. McCool, Aimy Sebastian, Gabriela G. Loots.

**Writing – review & editing:** Jillian L. McCool, Aimy Sebastian, Nicholas R. Hum, Blaine A. Christiansen, Gabriela G. Loots.

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
