## [Decision Letter · Decision Letter 0]

29 Apr 2024

PONE-D-24-11684CD206+Trem2+ Macrophage Accumulation in the Murine Knee Joint After Injury is Associated with Protection Against Post-Traumatic Osteoarthritis in MRL/MpJ Mice

PLOS ONE

Dear Dr. Loots,

Thank you for submitting your manuscript to PLOS ONE. After careful consideration, we feel that it has merit but does not fully meet PLOS ONE’s publication criteria as it currently stands. Therefore, we invite you to submit a revised version of the manuscript that addresses the points raised during the review process.

We look forward to receiving your revised manuscript.

Kind regards,

Zhiwen Luo

Academic Editor

PLOS ONE

Journal Requirements:

"This work was performed under the auspices of the U.S. Department of Energy by Lawrence Livermore National Laboratory under Contract DE-AC52-07NA27344. JLM, AS, NRH, DKM and SPW were supported by Lawrence Livermore National Laboratory LDRD 20-LW-002. AS, DKM, NRH, SPW, BA and GGL were supported by Department of Defense Awards PR180268 and PR192271. BAC was supported by the National Institute of Health grant R01 AR075013 and Department of Defense Award PR180268P1.

Author’s Role: Study design: JLM, AS and GGL; Data acquisition: JLM, DKM, CM, NRH, BA, and BAC. Data analysis and interpretation: JLM, AS, OAD, SPW, BAC, and GGL. JLM, AS and GGL wrote the manuscript.

"Department of Defense Awards PR180268,  PR180268P1 and PR192271.

National Institute of Health grant R01 AR075013."

"Department of Defense Awards PR180268, PR180268P1 and PR192271.

National Institute of Health grant R01 AR075013."

3.Thank you for stating the following financial disclosure:

 "Department of Defense Awards PR180268, PR180268P1 and PR192271.

National Institute of Health grant R01 AR075013."           

Please include this amended Role of Funder statement in your cover letter; we will change the online submission form on your behalf. "             

Additional Editor Comments :

Authors should respond to reviewer comments point by point.

Reviewers' comments:

Reviewer's Responses to Questions

**Comments to the Author**

1. Is the manuscript technically sound, and do the data support the conclusions?

Reviewer #1: Yes

Reviewer #2: Partly

2. Has the statistical analysis been performed appropriately and rigorously? 

Reviewer #1: Yes

Reviewer #2: Yes

3. Have the authors made all data underlying the findings in their manuscript fully available?

Reviewer #1: Yes

Reviewer #2: No

4. Is the manuscript presented in an intelligible fashion and written in standard English?

Reviewer #1: Yes

Reviewer #2: Yes

5. Review Comments to the Author

Reviewer #1: 1. Provide specific details regarding the nature of the joint injuries induced in the mice and how they correlate with human PTOA.

2. Discuss the potential limitations or caveats of using animal models to study a human disease, such as differences in the immune system and joint physiology.

3. Include information about the synovial joints that were analyzed and their role in PTOA pathogenesis, as well as any technical details regarding the single-cell RNA sequencing methodology.

4. Discuss the potential therapeutic implications of targeting the identified immune cell populations and molecular pathways in the treatment or prevention of PTOA.

Reviewer #2: Dear Author,

Thank you for submitting your article on the molecular programs contributing to the development or resolution of post-traumatic osteoarthritis (PTOA). After careful evaluation, we regret to inform you that we are unable to accept your manuscript for publication in our journal. We have identified several concerns that need to be addressed before reconsideration:

Limited Novelty: While the study of PTOA and its molecular mechanisms is of interest, the presented findings do not provide significant new insights beyond what has already been reported in the literature.

Insufficient Contextualization: The introduction does not adequately establish the importance and relevance of PTOA in the field of medicine. It is crucial to provide a comprehensive overview of the prevalence, clinical implications, and current understanding of PTOA.

Lack of Comparative Analysis: The article does not compare the findings of the study with previous studies or existing knowledge in the field. A more thorough comparison would help to assess the significance and contribution of the current study.

Incomplete Methodological Description: The article lacks a detailed description of the methods used for single-cell RNA sequencing and the analysis of immune cell populations. It is important to provide sufficient information on the experimental design, data analysis pipelines, and statistical approaches employed to ensure the validity and reproducibility of the results.

Inadequate Discussion of Functional Implications: The article does not sufficiently discuss the functional implications of the identified differences in monocyte and macrophage subpopulations between PTOA-susceptible and PTOA-resistant mice. A more comprehensive discussion of the functional aspects would enhance the understanding of the findings.

Insufficient Literature Review: The article lacks a thorough review of the existing literature on PTOA, including the molecular mechanisms involved in its development or resolution. A comprehensive review would provide a broader context for the presented findings.

Lack of Clarity in Results Presentation: The presentation of results and findings in the article is unclear and lacks proper organization. The information provided should be presented in a logical and concise manner to facilitate understanding.

Incomplete Discussion of Limitations: The article does not adequately address the limitations of the study, such as sample size, potential biases, or the generalizability of the findings. It is important to acknowledge and discuss these limitations to provide a balanced interpretation of the results.

6. PLOS authors have the option to publish the peer review history of their article (what does this mean?). If published, this will include your full peer review and any attached files.

Reviewer #1: No

Reviewer #2: No

---

## [Author Response · Author response to Decision Letter 0]

3 Jun 2024

Response to Reviewers

Reviewer #1: 

We kindly thank the reviewer for their suggestions and critiques and have made the following revisions and comments in response to their suggested points:

1. Provide specific details regarding the nature of the joint injuries induced in the mice and how they correlate with human PTOA.

We used a noninvasive tibial compression injury model that that mimics ACL rupture and subsequent PTOA development in human without surgical methodologies. This information has been added to the introduction lines 105-106. Over 15 studies have been published using this model and the originally published work as well as few other relevant studies have been cited. Our methodology section “Experimental Animals and ACL Injury Model” also briefly discusses the tibial compression injury model that was used in this study. 

2. Discuss the potential limitations or caveats of using animal models to study a human disease, such as differences in the immune system and joint physiology.

Thank you for the suggested revision. We have added additional information on limitations of our study to the discussion lines 677-693.

3. Include information about the synovial joints that were analyzed and their role in PTOA pathogenesis, as well as any technical details regarding the single-cell RNA sequencing methodology.

Immune (CD45+) cells from injured and uninjured knee joints of MRL and BL6 were analyzed in this study. Using a non-invasive tibial compression injury model, we induced ACL rupture in these mice to investigate PTOA pathogenesis. Technical details of the timepoints post injury, strain, age, gender of mice is discussed in our methodology section “Single-cell RNA sequencing and data analysis”. In this section, we have included methods for tissue collection as well as joint dissociation for synovial cell collection. Additionally, this section includes in depth descriptions of sequencing protocols and analysis of the data files. 

4. Discuss the potential therapeutic implications of targeting the identified immune cell populations and molecular pathways in the treatment or prevention of PTOA.

We have included additional information on potential therapeutic implications to the discussion and conclusion sections of the manuscript lines 658-662.

Reviewer #2: 

We kindly thank the reviewer for their careful review of this manuscript. We have included revisions to the manuscript and comments to address the following points: 

1. Limited Novelty: While the study of PTOA and its molecular mechanisms is of interest, the presented findings do not provide significant new insights beyond what has already been reported in the literature.

The enhanced healing capabilities of MRLs have been previously explored at the whole transcriptome and histopathology levels in several tissues (we have referenced Heydemann, Fitzgerald, Clark, Kwiatkowski, Rajnoch, Ward). However, no prior studies have investigated local changes to the immune response in MRL that could have contributed to the PTOA resistant phenotype, at a temporal and single-cell resolution as we have done in our study. Most current literature on MRL/MpJ used invasive surgical models such as medial meniscus displacement (Deng 2019 - PMID 31042406) and intraarticular fracture (Ward 2008 - PMID 18311808) to study PTOA progression. These invasive surgeries activate the immune system at the surgical site and may mask the true immune changes associated with PTOA progression.

In this study, we used a human-relevant, non-surgical model of ACL rupture to study PTOA progression. Using single-cell RNA sequencing technology we were able to identify injury-induced changes in various immune cell types/subtypes at an unprecedented resolution and the timepoints at which these changes occur. Our study highlighted that MRLs have increased number of a specific type of anti-inflammatory macrophages expressing Trem2 which may have contributed to their PTOA-resistant phenotype. We have also identified several differentially expressed genes between immune cell populations within the MRL and B6 joints. The therapeutic potential of some of these genes are being investigated in ongoing studies. Additionally, we believe the single cell data generated here, while being descriptive, will greatly add to the current field of PTOA and will aid as a foundation for additional hypothesis generating work.

2. Insufficient Contextualization: The introduction does not adequately establish the importance and relevance of PTOA in the field of medicine. It is crucial to provide a comprehensive overview of the prevalence, clinical implications, and current understanding of PTOA.

We thank the reviewer for the suggestion. Additional information on PTOA (cause, statistics of PTOA, economic burden etc.) have been included in the introduction as follows:

“Osteoarthritis (OA) is a painful joint disease that affects over 300 million people world-wide and is a leading cause of disability [PMID: 33671471]. Post-traumatic osteoarthritis (PTOA) is a form of osteoarthritis that develops as a result of joint injuries such as anterior cruciate ligament (ACL) rupture and meniscal tear and accounts for ~12% of all OA cases [PMID: 33671471]. This disease has a high-cost burden to healthcare systems worldwide and in the United States alone exceeds $10 billion annually [PMID:27145096]. Currently, there are no approved therapies to prevent the chronic pain and joint dysfunction associated with PTOA. Therefore, more research is needed to understand the molecular mechanisms of PTOA pathogenesis and to identify new targets for therapeutic development.”

Additionally, we discuss the current understanding of PTOA onset and pathogenesis, as well as the current immune cell contribution to arthritis progression in other animal and injury models in the introduction.

3. Lack of Comparative Analysis: The article does not compare the findings of the study with previous studies or existing knowledge in the field. A more thorough comparison would help to assess the significance and contribution of the current study.

We thank the reviewer for the suggestion. Multiple studies have shown that MRL mice are resistant to PTOA. However, the cellular and molecular mechanisms contributing to this resistance phenotype is not fully understood. The goal of our study was to investigate immune-driven mechanisms of PTOA resistance in MRL by first identifying the changing population dynamics of the immune cells after injury in B6 and MRL joints. These changing populations could lead to identification of novel therapeutic targets for PTOA. Thus, we have added the following to the introduction. 

“While the ability of MRL to repair their knee joint has been histologically evaluated in several studies, primarily using invasive injury models such as medial meniscus surgery, intraarticular fracture and cartilage wounds in the trochlear groove [PMIDs: 37979958, 18311808, 18455447, 31042406], cellular and molecular interactions leading to this resistant phenotype have not yet been fully elucidated. Identifying the cellular and molecular mediators of PTOA resistance in MRL could open up new avenues for therapeutic development for PTOA. ...“ 

“To enhance our understanding of the role the immune system plays in joint repair or joint damage after injury, we employed single-cell RNA sequencing (scRNA-seq) analysis of CD45+ cells from injured and uninjured knee joints from PTOA-resistant MRL and vulnerable B6 mice by using a noninvasive tibial compression injury model that mimics ACL rupture and subsequent PTOA development in humans....”

4. Incomplete Methodological Description: The article lacks a detailed description of the methods used for single-cell RNA sequencing and the analysis of immune cell populations. It is important to provide sufficient information on the experimental design, data analysis pipelines, and statistical approaches employed to ensure the validity and reproducibility of the results.

Our methodology section cites our previously published methodology for immune cell isolation from the synovial knee joint as well as describes the methodology used for single cell sequencing and analysis in detail. Please refer to methodology section “Single-cell RNA sequencing and data analysis”. 

5. Inadequate Discussion of Functional Implications: The article does not sufficiently discuss the functional implications of the identified differences in monocyte and macrophage subpopulations between PTOA-susceptible and PTOA-resistant mice. A more comprehensive discussion of the functional aspects would enhance the understanding of the findings.

We have included additional information on potential functional implications of many of our findings such as increased presence of Trem2+ macrophages and increased expression of genes such as Glo1 and Vwa5a. Although some populations such as the Spp1+ macrophages have been described extensively in the context of lung infections and cancer, not much is known about their role in joint or bone tissues. Further studies are required to understand their role in PTOA.

5. Insufficient Literature Review: The article lacks a thorough review of the existing literature on PTOA, including the molecular mechanisms involved in its development or resolution. A comprehensive review would provide a broader context for the presented findings.

Currently there is very little understanding of the molecular programs associated with PTOA onset in both a research and clinical setting. We describe the current findings of the literature associated with the immune systems response to injury, as well as resident function of these cells in a homeostatic knee joint in the introduction. Additionally, current literature discussing potential roles of different immune cells and specific genes identified in this study are included in the discussion. The primary objective this study was to understand the cellular mechanisms that contribute to the PTOA resistant phenotype in MRL through identification of local cellular changes within the joints space after trauma. Additionally, we have reviewed and cited multiple publications investigating PTOA progression in MRL as well as the healing capacity of other tissues from MRL. 

6. Lack of Clarity in Results Presentation: The presentation of results and findings in the article is unclear and lacks proper organization. The information provided should be presented in a logical and concise manner to facilitate understanding.

We apologize for the lack of clarity. We have modified the introduction, discussion and conclusions significantly to facilitate the understanding of the relevance of our findings. We understand that single-cell data from longitudinal experiments such as ours can be overwhelming but, we believe the data generated here would add great value to the community and could be hypothesis generating for others working in the field. The results section provides a top-down view of the cell types present, starting with a broad characterization of the immune cell populations as a whole. Within the Results section, Sub-section 1 discusses the baseline populations present prior to injury and after injury. We then moved forward to discuss the largest population shifts seen in the data (neutrophils and mono-macs). Each of the following sub-sections is organized to further discuss specific subpopulations of immune cells with unique transcriptomic profiles in response to injury.

7. Incomplete Discussion of Limitations: The article does not adequately address the limitations of the study, such as sample size, potential biases, or the generalizability of the findings. It is important to acknowledge and discuss these limitations to provide a balanced interpretation of the results.

Thank you for the suggestion. We have added study limitations to the discussion section.

---

## [Decision Letter · Decision Letter 1]

18 Sep 2024

PONE-D-24-11684R1CD206+Trem2+ Macrophage Accumulation in the Murine Knee Joint After Injury is Associated with Protection Against Post-Traumatic Osteoarthritis in MRL/MpJ MicePLOS ONE

Dear Dr. Loots,

Thank you for submitting your manuscript to PLOS ONE. After careful consideration, we feel that it has merit but does not fully meet PLOS ONE’s publication criteria as it currently stands. Therefore, we invite you to submit a revised version of the manuscript that addresses the points raised during the review process.

**ACADEMIC EDITOR: **Thank you for submitting your manuscript to the Journal and as voucan see that the reviewer think your manuscript is interesting and provide valuable comments for you reference before publication. Please submit the revised manuscript ASAP and also include a rebuttal that would clearly list all the responses to the reviewer's comments.

We look forward to receiving your revised manuscript.

Kind regards,

Zhiwen Luo

Academic Editor

PLOS ONE

Journal Requirements:

Reviewers' comments:

Reviewer's Responses to Questions

**Comments to the Author**

1. If the authors have adequately addressed your comments raised in a previous round of review and you feel that this manuscript is now acceptable for publication, you may indicate that here to bypass the “Comments to the Author” section, enter your conflict of interest statement in the “Confidential to Editor” section, and submit your "Accept" recommendation.

Reviewer #3: (No Response)

Reviewer #4: All comments have been addressed

2. Is the manuscript technically sound, and do the data support the conclusions?

Reviewer #3: Yes

Reviewer #4: Yes

3. Has the statistical analysis been performed appropriately and rigorously? 

Reviewer #3: Yes

Reviewer #4: Yes

4. Have the authors made all data underlying the findings in their manuscript fully available?

Reviewer #3: (No Response)

Reviewer #4: Yes

5. Is the manuscript presented in an intelligible fashion and written in standard English?

Reviewer #3: (No Response)

Reviewer #4: Yes

6. Review Comments to the Author

Reviewer #3: This study mainly involved animal model construction and single cell sequencing analysis of synovial joint tissue. We suggest that more in vivo histological tests and potential mechanisms be explored.

Reviewer #4: Here, the authors have revised their manuscript focused on the differences between B6 mice (which are highly susceptible to PTOA) and MRL superhealer mice (which show protection), with respect to synovial immune cell populations. Overall I think this is a valuable contribution to the field given that our understanding of why some injured people go on to develop PTOA (often rapidly) and why some don't, is very limited. Studying natural differences in regenerative capacity within mice strains is a good approach for tackling this conundrum.

I believe that the authors have sufficiently addressed the comments from the first round of review, and improved the manuscript. As such, I recommend it for acceptance. Here I am providing some additional comments for the authors to take into consideration.

1) A major hurdle to studying intra-articular/synovial cells, especially immune cells, is harvesting truly representative "joint resident" cells and avoiding contamination from other sources - the most notorious being the bone marrow, given how dense with immune cells it is. Here the authors have painstakingly tried to overcome and address this hurdle, which I applaud them for. Having bona fide intra-articular cells in a single cell dataset is critical for drawing conclusions about the nuanced populations that reside in or infiltrate the joint space in homeostasis and after injury.

2) Specifically, relating to potential BM neutrophil contamination, line 382 states "to determine if neutrophils in our digests were bone marrow-derived..". In the future, it could be worth considering staining BM/circulating neutrophils prior in an inducible fashion (whether with a creERT2 or fluor-labeled in vivo delivery prior to harvest, to truly distinguish BM/circulating neutrophils from bona fide intra-articular cells. This is at the forefront of the field and would be a worthwhile pursuit, as it has not been done before (to my knowledge).

3) Minor suggestion for future studies - we and others have found that dramatically slowing the rate of perfusion helps to prevent bursting of delicate narrow capillary beds in peripheral tissues, like synovium. Here the authors perfused at 10mL/min - I would suggest slowing this to 200mL/h.

4) In lines 438-452, there may be some confusion about cluster numbers that you are referring to. Please double check this to ensure it's correct throughout this section.

7. PLOS authors have the option to publish the peer review history of their article (what does this mean?). If published, this will include your full peer review and any attached files.

Reviewer #3: No

Reviewer #4: **Yes: **Alexander Knights

---

## [Author Response · Author response to Decision Letter 1]

8 Oct 2024

Reviewer #3: This study mainly involved animal model construction and single cell sequencing analysis of synovial joint tissue. We suggest that more in vivo histological tests and potential mechanisms be explored.

We kindly thank the reviewer for their suggestion. Although this study mainly describes findings based on single cell sequencing analysis, we have included several flowcytometry and histology data verifying our findings (Figure 2 B-D, Figure 3 A-B and Figure 7). Indeed, additional mechanistic experiments are worthy of doing, however we believe the single cell data generated here would add great value to the community and could be hypothesis generating for others working in the field. We appreciate any feedback that can improve this work and make it useful to the community, but it is impossible for new mouse experiments to be initiated at this point due to lack of funding.

Reviewer #4: Here, the authors have revised their manuscript focused on the differences between B6 mice (which are highly susceptible to PTOA) and MRL superhealer mice (which show protection), with respect to synovial immune cell populations. Overall I think this is a valuable contribution to the field given that our understanding of why some injured people go on to develop PTOA (often rapidly) and why some don't, is very limited. Studying natural differences in regenerative capacity within mice strains is a good approach for tackling this conundrum.

I believe that the authors have sufficiently addressed the comments from the first round of review, and improved the manuscript. As such, I recommend it for acceptance. Here I am providing some additional comments for the authors to take into consideration.

1) A major hurdle to studying intra-articular/synovial cells, especially immune cells, is harvesting truly representative "joint resident" cells and avoiding contamination from other sources - the most notorious being the bone marrow, given how dense with immune cells it is. Here the authors have painstakingly tried to overcome and address this hurdle, which I applaud them for. Having bona fide intra-articular cells in a single cell dataset is critical for drawing conclusions about the nuanced populations that reside in or infiltrate the joint space in homeostasis and after injury.

We kindly thank the reviewer for their positive feedback.

2) Specifically, relating to potential BM neutrophil contamination, line 382 states "to determine if neutrophils in our digests were bone marrow-derived..". In the future, it could be worth considering staining BM/circulating neutrophils prior in an inducible fashion (whether with a creERT2 or fluor-labeled in vivo delivery prior to harvest, to truly distinguish BM/circulating neutrophils from bona fide intra-articular cells. This is at the forefront of the field and would be a worthwhile pursuit, as it has not been done before (to my knowledge).

We kindly thank the reviewer for their suggestion. In future studies we will try to better label neutrophils to truly distinguish BM/circulating neutrophils from bona fide intra-articular cells, this is a current limitation and we have stated this in the discussion now.

‘Furthermore, we currently can’t distinguish between BM/circulating neutrophils from neutrophils that have infiltrated the joint tissue, but future lineage tracing experiments will aid in this endeavor.’

3) Minor suggestion for future studies - we and others have found that dramatically slowing the rate of perfusion helps to prevent bursting of delicate narrow capillary beds in peripheral tissues, like synovium. Here the authors perfused at 10mL/min - I would suggest slowing this to 200mL/h.

We kindly thank the reviewer for their suggestion. In future studies, we will try slowing down the rate of perfusion. 

4) In lines 438-452, there may be some confusion about cluster numbers that you are referring to. Please double check this to ensure it's correct throughout this section.

We kindly thank the reviewer for pointing out this inconsistency. We have adjusted the cluster number for Spp1+Cav1+ subpopulation (changed cluster 2 to cluster 3) in this section of the manuscript. We have also reviewed the remainder of the text and ensured there are no more misnumbered clusters.

---

## [Editor Report · Decision Letter 2]

10 Oct 2024

CD206+Trem2+ Macrophage Accumulation in the Murine Knee Joint After Injury is Associated with Protection Against Post-Traumatic Osteoarthritis in MRL/MpJ Mice

PONE-D-24-11684R2

Dear Dr. Loots,

We’re pleased to inform you that your manuscript has been judged scientifically suitable for publication and will be formally accepted for publication once it meets all outstanding technical requirements.

Kind regards,

Zhiwen Luo

Academic Editor

PLOS ONE
---

## [Editor Report · Acceptance letter]

16 Dec 2024

PONE-D-24-11684R2 

PLOS ONE

Dear Dr. Loots, 

I'm pleased to inform you that your manuscript has been deemed suitable for publication in PLOS ONE. Congratulations! Your manuscript is now being handed over to our production team.

Kind regards, 

on behalf of

Dr. Zhiwen Luo 

Academic Editor

PLOS ONE